# NegMerge: Sign-Consensual Weight Merging for Machine Unlearning

**Hyo Seo Kim** [1][⋆]  **Dongyoon Han** [2][†]  **Junsuk Choe** [1][†]

## Abstract

Machine unlearning aims to selectively remove specific knowledge from a trained model. Existing approaches, such as Task Arithmetic, fine-tune the model on the forget set to create a task vector (*i.e.*, a direction in weight space) for subtraction from the original model's weight. However, their effectiveness is highly sensitive to hyperparameter selection, requiring extensive validation to identify the optimal vector from many fine-tuned candidates. In this paper, we propose a novel method that utilizes all fine-tuned models trained with varying hyperparameters instead of a single selection. Specifically, we aggregate the computed task vectors by retaining only the elements with consistent shared signs. The merged task vector is then negated to induce unlearning on the original model. Evaluations on zero-shot and standard image recognition tasks across twelve datasets and four backbone architectures show that our approach outperforms state-of-the-art methods while requiring similar or fewer computational resources. Code is available at https://github.com/naver-ai/negmerge.

## 1. Introduction

The *Right to be Forgotten* regulation (Hoofnagle et al., 2019) allows individuals to request the deletion of their personal data. However, applying this concept to machine learning models is challenging because the training process deeply embeds the data into the model's parameters, making it difficult to remove its influence. The most straightforward solution is to remove the data from the training set and retrain the model from scratch, which requires enormous computational resources. As a result, ensuring that models selectively forget specific learned patterns becomes a challenging task. *Machine unlearning* (Koh & Liang, 2017; Golatkar et al., 2020; Thudi et al., 2022) offers a solution by enabling models to erase specific knowledge without the need for full retraining.

Despite promising results, many existing methods struggle to remove only the target knowledge while preserving the rest. This challenge arises because fine-tuning, in its effort to erase knowledge from the *forget set* (i.e., the data to be forgotten), often disrupts the knowledge preserved in the *retain set* (i.e., the remaining data) (Chen et al., 2023; Fan et al., 2023). A known method robust to this issue is Task Arithmetic (Ilharco et al., 2022a), which avoids directly fine-tuning the original model. It calculates a task vector by determining the parameter-wise difference between the original model and a separately fine-tuned model on the forget set. The task vector is then subtracted from the original model. This process, referred to as *forgetting by negation*, has demonstrated strong unlearning performance while preserving the model's knowledge.

However, our findings reveal that achieving effective unlearning performance with this method requires careful hyperparameter selection through a *validation* process. This process is both time-consuming and computationally expensive, as it involves evaluating multiple fine-tuned models trained with different hyperparameters. Moreover, we argue that selecting only a single model, as in existing methods (Ilharco et al., 2022a; Ortiz-Jimenez et al., 2024), does not guarantee optimal performance. To address this, we propose leveraging all candidate models to effectively utilize the information they contain, rather than selecting just one and discarding the rest.

In this paper, we present NegMerge, a novel method that enhances the process of forgetting by negation. Our approach computes a final task vector by merging task vectors derived from multiple fine-tuned models. During this merging process, we preserve elements with consistent signs across the task vectors and mask out elements with inconsistent signs, setting them to zero. The subsequent steps align with the standard forgetting by negation process, where the final task vector is subtracted from the original model to induce forgetting (Ilharco et al., 2022a). Our method is inspired by model merging techniques (Wortsman et al., 2022; Yang et al., 2023; Jang et al., 2024), which utilize

---
[⋆]Work done during an internship at NAVER AI Lab. [†]Equal advising. [1]Sogang University [2]NAVER AI Lab. Correspondence to: Junsuk Choe <jschoe@sogang.ac.kr>, Dongyoon Han <dongyoon.han@navercorp.com>.

*Proceedings of the 42nd International Conference on Machine Learning*, Vancouver, Canada. PMLR 267, 2025. Copyright 2025 by the author(s).

multiple fine-tuned models produced during the validation process. Building on this concept, we adapt it to the context of machine unlearning.

We demonstrate the effectiveness of our method in two experimental settings. The first setting aims to make a classification model with zero-shot recognition capabilities, such as CLIP (Radford et al., 2021), unable to recognize specific knowledge. The second setting focuses on removing knowledge associated with a specific subset of the training data in a standard image classification model (Chen et al., 2023; Fan et al., 2023). We validate our method on ViT (Dosovitskiy et al., 2021), ResNet (He et al., 2016), VGG (Simonyan, 2014), and Swin-T (Liu et al., 2021) architectures across a total of 12 datasets. Our approach achieves new state-of-the-art performance while utilizing computational resources that are comparable to or fewer than those of existing methods.

## 2. Related work

**Machine Unlearning for Image Classification.** Existing methods have been applied mainly to two tasks. The first focuses on reducing zero-shot recognition performance for specific knowledge in vision-language models, such as CLIP (Ilharco et al., 2022a; Ortiz-Jimenez et al., 2024). The second involves unlearning knowledge tied to a subset of the training data in standard image classification models (Chen et al., 2023; Fan et al., 2023). These two tasks have traditionally been treated as separate research areas.

In the first task, the negation method proposed in Task Arithmetic (Ilharco et al., 2022a) is applied for unlearning. More recently, a linear negation method based on the Neural Tangent Kernel (Jacot et al., 2018; Ortiz-Jimenez et al., 2024) has been introduced. This method enables arithmetic in the linear space by fine-tuning the model in the tangent space. Both approaches depend on a single fine-tuned model to calculate task vectors, selecting the single best model from numerous fine-tuned models derived during the validation process.

Machine unlearning for a standard image classifier usually involves fine-tuning the original model. Fine-tuning (Warnecke et al., 2021) and $\ell_1$-sparse (Jia et al., 2023) aim to overfit the model only on the retain set to erase the knowledge of the forget set. Influence (Koh & Liang, 2017) and SalUn (Fan et al., 2023) utilize both the retain and forget sets to selectively degrade performance on the forget set while maintaining it on the retain set.

In many cases, the size of the forget set is significantly smaller than that of the retain set. In such scenarios, machine unlearning methods that require the retain set can be inefficient. This challenge has driven the development of approaches that perform unlearning using only the forget set. Existing methods (Golatkar et al., 2020; Chen et al., 2023)

attempt to induce forgetting by relabeling the forget set to different classes and fine-tuning the model accordingly. However, they often suffer from catastrophic forgetting, as the retain set is not used during fine-tuning, leading to the loss of its knowledge.

We propose a novel approach based on Task Arithmetic (Ilharco et al., 2022a) that leverages multiple fine-tuned models to tackle the aforementioned challenges. By incorporating insights from these models, our method computes a more effective task vector, enhancing unlearning performance while preserving retain set knowledge. Notably, our approach requires only the forget set and is effective in both image classification tasks.

**Model Merging.** Model Soups (Wortsman et al., 2022) addresses the inefficiency of discarding many models during the validation process, where only a single best model is selected. They argue that merging the weights of all generated models can improve generalization performance without additional computational overhead. Task Arithmetic (Ilharco et al., 2022a) introduces task vectors, showing that desired knowledge can be effectively added to or removed from models through simple arithmetic operations with these vectors. AdaMerging (Yang et al., 2023) autonomously learns model merging coefficients at the task or layer level and performs this process without relying on the original training data. TIES-Merging (Yadav et al., 2024) incorporates a trimming stage that retains only elements with large magnitudes during the merging process and resolves sign conflicts between elements through a voting mechanism, merging only elements corresponding to the selected sign. MagMax (Marczak et al., 2024) selects and merges only the elements with the largest magnitude from the task vectors. Several approaches further improve merging methods by localizing task-relevant parameters. Skill Localization (Panigrahi et al., 2023) selects parameters with the largest changes during fine-tuning, Localize-and-Stitch (He et al., 2024) retains the top-k% parameters based on magnitude, and TALL Mask (Wang et al., 2024) masks out parameters with small magnitudes.

Our proposed technique merges task vectors by selecting only elements with the same sign while masking the remaining elements with zero. This approach has not been explored in existing methods, and our evaluation shows that it is highly effective in machine unlearning.

## 3. Method

### 3.1. Background

**Task Arithmetic** involves defining a *task vector* $\tau_t$, by subtracting the parameters of a pre-trained model $\theta_{pre}$ from those of a fine-tuned model $\theta_{ft}^t$ for a specific target task $t$: $\tau_t = \theta_{ft}^t - \theta_{pre}$. For handling multiple tasks simultane-

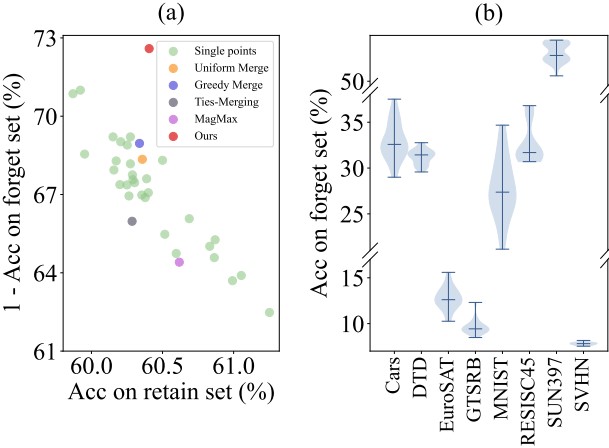

*Figure 1.* **Hyperparameter Sensitivity in Negation Methods.** Each point in (a) represents the accuracy on the forget and retain sets, with the error rate (calculated as $1 -$ accuracy) used for the forget set to improve visibility. Green points indicate results from models fine-tuned with various hyperparameter settings, while points in other colors show results from different methods. Ours breaks the trade-off between unlearning and retaining performance. Panel (b) presents the accuracy distribution on the forget set across different hyperparameter choices, which vary by up to 15 percentage points. Both (a) and (b) are based on the CLIP ViT-B/32 model, where (a) focuses on the Cars dataset, and (b) includes experiments across eight datasets.

ously, task vectors from individual tasks can be combined as $\tau = \sum_t \tau_t$. This combined vector enables the model to be adjusted in a desired direction by modifying the original model's weights. The updated model weights are computed as: $\theta_{new} = \theta_{pre} \pm \lambda\tau$, where $\lambda$ is a scaling hyperparameter that controls the magnitude of adjustment.

A key application of Task Arithmetic (Ilharco et al., 2022a) is to make a model forget certain capabilities. This can be achieved through the *negation* of a task vector from the original weight, which decreases performance on a target task. For example, Task Arithmetic can be applied to unlearning in models such as CLIP (Radford et al., 2021), which is known for its strong zero-shot recognition capabilities. Task Arithmetic demonstrated that a task vector derived from a CLIP model fine-tuned on a specific dataset (*e.g.*, Cars) can reduce the model's accuracy on that dataset while maintaining its overall accuracy on a general dataset (*e.g.*, ImageNet). While Task Arithmetic has shown promise in machine unlearning, limited research has explored how to compute a task vector specifically optimized for unlearning. Our research aims to address this gap.

**Motivation.** Through our pilot study, we identified two major challenges in obtaining effective task vectors for unlearning. First, it is challenging to balance reducing accuracy on the forget set while maintaining accuracy on the retain set. As shown in Figure 1 (a), hyperparameter sets

that preserve retain set performance tend to exhibit poor unlearning performance, and vice versa. To address this, we propose a method that leverages multiple models to combine their strengths, which a single model alone cannot achieve. Figure 1 (a) shows that our method overcomes this trade-off. In contrast, existing merging methods such as Uniform Merge and Greedy Merge (Wortsman et al., 2022), TIES-Merging (Yadav et al., 2024), and MagMax (Marczak et al., 2024) struggle with this trade-off, highlighting the need for a merging strategy designed for unlearning. The full results for all eight datasets are provided in Figure B1.

Second, unlearning performance is highly sensitive to the hyperparameter settings used during fine-tuning. As shown in Figure 1 (b), accuracy on the forget set can vary by up to 15 percentage points depending on the hyperparameters. This sensitivity not only affects performance stability but also makes the tuning process time-consuming and computationally expensive, as it requires additional tuning of the scaling hyperparameter $\lambda$ for each fine-tuned model. However, our method mitigates the sensitivity by merging all fine-tuned models obtained from diverse hyperparameter configurations. This strategy is inspired by Model Soups (Wortsman et al., 2022), which shows that averaging the weights of independently fine-tuned models can improve performance and robustness. The key idea is that different hyperparameter choices introduce independent variations, and merging helps cancel out noise from any single run. As a result, Figure 1 (a) shows that our method achieves stronger unlearning performance, while significantly reducing the time needed for scaling hyperparameter tuning by merging models derived during validation. Discussions of the computational costs are described in Section 3.3.

### 3.2. The Proposed Method: `NegMerge`

We propose a method to achieve effective unlearning by integrating task vectors, given that multiple models are fine-tuned on the forget set under various training configurations. A detailed description of each step is provided below, and Figure 2 illustrates the overview of our method.

**Step 1) Calculating Diverse Task Vectors.** As previously mentioned, machine unlearning based on Task Arithmetic (Ilharco et al., 2022a) is highly sensitive to hyperparameters, making it essential to identify optimal hyperparameters through validation. In standard validation processes, various training configurations are employed, such as adjusting hyperparameters like the learning rate or applying additional data augmentation techniques, to derive multiple models. Our study emulates such validation procedures to construct a model pool, which is then utilized to calculate diverse task vectors. Importantly, we restrict the number of models to the typical range used in hyperparameter validation (10 to 30 models), ensuring that the use of multiple

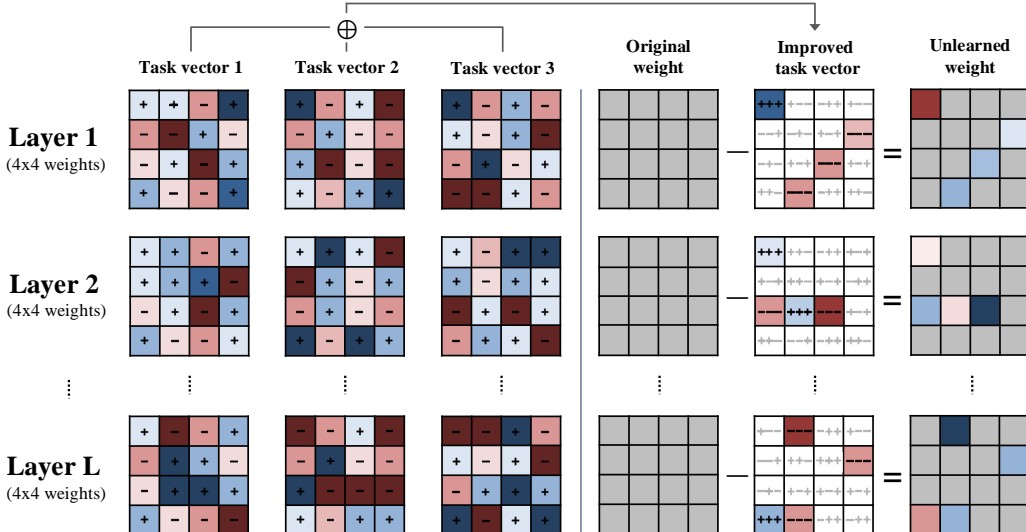

*Figure 2.* **Illustration of the proposed method.** `NegMerge` enhances Task Arithmetic by computing an improved task vector. Specifically, 1) task vectors derived from multiple fine-tuned models trained with different hyperparameters are utilized. 2) We compute the improved task vector by merging ($\oplus$) only the elements that retain a consistent sign across task vectors while masking elements with differing signs to zero. 3) This refined task vector is used for negation from the original model. The color intensity in the cells reflects the magnitude of the task vector elements; darker blue represents larger positive values, lighter blue indicates smaller positives, while darker red represents larger negative values, and lighter red indicates smaller negatives.

models does not introduce additional computational overhead. Detailed information on this approach is provided in Section 4.1.

**Step 2) Identifying Task Vector Elements for Forget Set.** After deriving task vectors from the fine-tuned models, we analyze them to determine which elements correspond to the forget set. We conjecture that elements that consistently show the same sign across task vectors are attributed to the forget set, as each model is trained to align with this set, regardless of the training configurations. On the other hand, components that exhibit inconsistent signs are considered less related to the forget set, as their variations are more likely a result of different training configurations rather than supervision from the forget set. Our conjecture regarding sign conflicts is supported by the unlearning performance reported in Table 3 and qualitative results in Figure 3.

**Step 3) Final Task Vector for Negation.** We compute the final task vector using the following formulation:

$$\tau_{\text{merged}} = \frac{1}{n} \sum_{k=1}^{n} \left( \tau_k \odot \mathbf{1}_{\text{sign-consistent}} \right), \quad (1)$$

where $n$ is the number of task vectors, $\odot$ denotes the Hadamard product (element-wise multiplication), and the vector $\mathbf{1}_{\text{sign-consistent}}$ acts like a filter, containing 1 for elements where the signs of the corresponding components across all task vectors $\tau_k$ are the same and 0 where the signs differ[1]. As a result, only the components with consistent

signs across all task vectors contribute to the final task vector, while those with differing signs are excluded by being set to zero. We then perform unlearning by negating this final task vector to the original model (Ilharco et al., 2022a). Theoretical analysis is provided in Appendix E.

### 3.3. Analysis on Computational Cost

We analyze the computational cost from four perspectives: inference time complexity, merge time, storage, and runtime memory. All compared methods, including Task Arithmetic (Ilharco et al., 2022a), share the same model pool, ensuring no additional computational overhead from using multiple models. To simulate a realistic validation process, we vary the number of fine-tuned models ($n$) between 10 and 30. As shown in Table 4, our method consistently demonstrates robustness to the number of fine-tuned models.

**Inference Time Complexity.** The standard setup for task vector negation-based methods (Ilharco et al., 2022a; Ortiz-Jimenez et al., 2024) involves performing inference with $m$ different scaling coefficients $\lambda$ for each of the $n$ fine-tuned models. Specifically, this process requires $m = 20$ inferences per model in prior works (Ilharco et al., 2022a; Ortiz-Jimenez et al., 2024), resulting in a significant computational cost of $O(mn)$. In contrast, our method performs inference only $m$ times on a single merged task vector, reducing the computational cost to $O(m)$. While Task Arith-

---

[1]This operation is based on sign unanimity and could be adjusted with additional hyperparameters to allow partial consensus; however, we opt for a simpler approach.

metic employs a single model during its final stage, achieving optimal performance with this approach entails greater computational demands compared to our method. This demonstrates the computational advantages of our approach over competing methods.

**Merge Time.** Our method requires checking the sign of elements across task vectors, making it slower than simpler approaches such as MagMax (Marczak et al., 2024), which identifies only the maximum values, or Uniform Merge (Wortsman et al., 2022), which averages task vectors. However, it is much faster than methods that rely on more complex operations, such as TIES-Merging (Yadav et al., 2024) or Greedy Merge (Wortsman et al., 2022). Detailed comparisons are provided in Table 1.

**Storage.** Our approach might initially seem storage-intensive, as it involves storing all fine-tuned models for merging. However, this is not the case in practice. Rather than saving every fine-tuned model, we dynamically update a mask, $\mathbf{1}_{\text{sign-consistent}}$, based on the sign consensus of each element when a new model is trained. This eliminates the need to store all fine-tuned models. Consequently, the proposed technique achieves more effective unlearning while maintaining the same storage requirements as traditional single-model-based methods. By contrast, methods like TIES-Merging cannot utilize such a mechanism, further underscoring the advantages of our approach.

**Runtime Memory.** Our method provides a clear advantage in runtime memory efficiency. As demonstrated in Table 4, a large proportion of the weights in the task vector are zeroed out during the merging process, with only 5–10% of the total weight elements remaining active. This significant sparsity enables lightweight memory management techniques, like weight lookup tables, which store only the active weights and further reduce runtime memory consumption. As a result, our method achieves superior efficiency compared to baseline approaches.

## 4. Experiments

### 4.1. Experimental Setups

**Two Unlearning Scenarios.** We examine two distinct unlearning scenarios. The first involves a vision-language model, such as CLIP (Radford et al., 2021), being made to forget the knowledge associated with a specific dataset. The second scenario focuses on a standard image classification network trained with cross-entropy loss, where the model is instructed to forget knowledge of particular data points within its training set. The specific implementation details for each scenario are provided in Appendix A.

**Datasets and Backbones.** In the CLIP scenario, we follow the training and evaluation protocols from the Task Arithmetic paper (Ilharco et al., 2022a). We assess unlearning performance on eight datasets: SUN397 (Xiao et al., 2016), Cars (Krause et al., 2013), RESISC45 (Cheng et al., 2017), EuroSAT (Helber et al., 2019), SVHN (Yuval, 2011), GT-SRB (Stallkamp et al., 2011), MNIST (LeCun, 1998), and DTD (Cimpoi et al., 2014), while using ImageNet (Deng et al., 2009) as the retain set to evaluate retaining performance. All experiments are conducted using pre-trained CLIP ViT-{B/32, B/16, L/14} models (Radford et al., 2021). In the standard classifier scenario, we evaluate unlearning performance on CIFAR-10 (Krizhevsky et al., 2009), CUB-200-2011 (Wah et al., 2011), and Tiny ImageNet (Le & Yang, 2015) using ResNet-18 (He et al., 2016), VGG-16 (Simonyan, 2014), and Swin-T (Liu et al., 2021) models.

**Baselines and Metrics.** For the CLIP scenario, we compare our method with five existing methods: Task Arithmetic (Ilharco et al., 2022a), Uniform Merge and Greedy Merge (Wortsman et al., 2022), TIES-Merging (Yadav et al., 2024), and MagMax (Marczak et al., 2024). For Greedy Merge, we rank models by their loss on the retain set and merge them in a direction that minimizes this loss. We evaluate performance by measuring accuracies on the forget set $D_f$ and the retain set $D_r$. In the standard classifier scenario, we follow SalUn (Fan et al., 2023) to compare our method against eight unlearning techniques: Fine-tuning (Warnecke et al., 2021), Random Labeling (Golatkar et al., 2020), Gradient Ascent (Thudi et al., 2022), Influence Unlearning (Koh & Liang, 2017), $\ell_1$-sparse (Jia et al., 2023), Boundary Shrink and Expand (Chen et al., 2023), and SalUn. In addition, we include merging-based baselines Task Arithmetic, Uniform Merge, TIES-Merging, and MagMax. Greedy Merge is infeasible for comparison in this scenario when using only the forget set. The objective is to match the unlearned model's performance to that of a fully retrained model. We use the accuracies of the retain set $D_r$, forget set $D_f$, and test set $D_{test}$ to evaluate performance. To assess privacy protection, we employ the Membership Inference Attack (MIA) metric (Carlini et al., 2022), following the MIA-Efficacy metric from (Fan et al., 2023; Jia et al., 2023). Higher MIA-Efficacy implies more unlearning, as it measures how much less information the model retains about the forget data.

### 4.2. Experimental Results

**CLIP Unlearning Scenario.** Table 1 presents the evaluation results across three CLIP models (ViT-B/32, ViT-B/16, and ViT-L/14). Note that the retain set ($D_r$) accuracies for all methods remain around 60%, as we follow Task Arithmetic (Ilharco et al., 2022a) to ensure the model retains at least 95% of the pre-trained model's original accuracy (66.66%) on the validation set. This allows for a direct comparison of forget set accuracies.

Our method achieves the best reduction in accuracy on the

*Table 1.* **Unlearning Performance on CLIP ViT Models.** Results are shown for CLIP ViT-{B/32, B/16, L/14}, reporting average accuracy (%) on the eight datasets we wish to forget (Cars, DTD, EuroSAT, GTSRB, MNIST, RESISC45, SUN397, and SVHN), and the general dataset to retain (ImageNet). [*] indicates that the numbers are borrowed from the original papers. [†] denotes the best performance achieved through hyperparameter search. [‡] combines models in descending order of losses. Time denotes the merging time, measured in seconds, taken to merge 30 models on the Cars dataset using CLIP ViT-B/32, which is averaged over three runs. NegMerge consistently achieves the lowest forget set accuracy across all backbones, indicating strong unlearning performance.

| Method | ViT-B/32 | | ViT-B/16 | | ViT-L/14 | | Time (sec) |
|---|---|---|---|---|---|---|---|
| | Acc $D_f(\downarrow)$ | Acc $D_r$ | Acc $D_f(\downarrow)$ | Acc $D_r$ | Acc $D_f(\downarrow)$ | Acc $D_r$ | |
| Pre-trained | 48.13 | 63.33 | 55.49 | 68.32 | 65.19 | 75.54 | - |
| Task Arithmetic | | | | | | | |
|   *Paper number*[*] | 24.00 | 60.90 | 21.30 | 65.40 | 19.00 | 72.90 | - |
|   Single Best Model[†] | 23.63 | 60.60 | 20.64 | 64.04 | 19.17 | 72.09 | - |
|   Uniform Merge | 22.50 | 60.55 | 21.51 | 64.60 | 18.10 | 71.91 | $12_{\pm 0.1}$ |
|   Greedy Merge[‡] | 23.31 | 60.75 | 21.34 | 64.54 | 17.71 | 71.99 | $607_{\pm 2.6}$ |
|   TIES-Merging | 26.21 | 61.08 | 23.78 | 64.72 | 22.70 | 72.41 | $128_{\pm 10.1}$ |
|   MagMax | 25.24 | 60.95 | 24.45 | 64.78 | 21.71 | 72.55 | $24_{\pm 1.8}$ |
|   **NegMerge (ours)** | **20.76** | 60.36 | **19.24** | 64.54 | **17.32** | 72.08 | $37_{\pm 1.2}$ |
| Linear Task Arithmetic | | | | | | | |
|   *Paper number*[*] | 10.90 | 60.80 | 11.30 | 64.80 | - | - | - |
|   Single Best Model[†] | 8.88 | 60.16 | 6.92 | 64.62 | - | - | - |
|   Uniform Merge | 9.12 | 60.47 | 6.84 | 65.26 | - | - | $19_{\pm 2.3}$ |
|   Greedy Merge[‡] | 8.73 | 60.27 | 6.80 | 64.72 | - | - | $1696_{\pm 35.3}$ |
|   TIES-Merging | 10.66 | 60.38 | 8.44 | 65.12 | - | - | $378_{\pm 8.0}$ |
|   MagMax | 11.33 | 60.67 | 8.65 | 65.17 | - | - | $164_{\pm 2.4}$ |
|   **NegMerge (ours)** | **8.03** | 60.58 | **6.60** | 65.40 | - | - | $194_{\pm 1.6}$ |

forget set $D_f$ across all backbones, which demonstrates its generalizability regardless of model size. Specifically, for CLIP ViT-B/32, our method reduces the forget set ($D_f$) accuracy to 20.76%, outperforming Task Arithmetic (23.63%), Uniform Merge (22.50%), and Greedy Merge (23.31%). Our method maintains strong performance across different model variants. For CLIP ViT-B/16, it reduces forget set accuracy to 19.24%, outperforming Task Arithmetic (20.64%). Similarly, for CLIP ViT-L/14, our approach achieves the best forget set performance, lowering accuracy to 17.32%. In contrast, MagMax and TIES-Merging exhibit weaker unlearning performance.

Regarding merging time, our method requires slightly more time than Uniform Merge and MagMax but offers significantly better effectiveness. Furthermore, while Greedy Merge (Wortsman et al., 2022) and TIES-Merging (Yadav et al., 2024) are considerably slower, our method outperforms them by a large margin in accuracy.

To provide a more comprehensive evaluation of our method, we employ *Linear Task Arithmetic*, where Neural Tangent Kernel (NTK) (Jacot et al., 2018; Ortiz-Jimenez et al., 2024) is applied to the standard Task Arithmetic (Ilharco et al., 2022a). The experimental results are presented in the lower part of Table 1, where we conduct evaluations using the CLIP ViT-B/32 and ViT-B/16 backbones. Due to computational resource constraints, we are unable to include results

for ViT-L/14. Our method achieves the best unlearning performance, while the second-best method, Greedy Merge, requires significantly more time for merging (1696.5 and 194.2, respectively).

Full results for all eight datasets and three CLIP models are provided in Appendix B.1.

**Standard Classifier Unlearning Scenario.** Table 2 presents a comparison of various unlearning techniques on CIFAR-10 using ResNet-18. In this task, we randomly select 10% of the training set as the forget set. The goal is to make the model forget the knowledge associated with the forget set while maintaining its performance on the retain set. The fully retrained model serves as the ideal benchmark for forget, retain, and privacy tasks. Following SalUn (Fan et al., 2023), we report the Avg. Gap metric to evaluate how closely each unlearning method replicates the performance of the retrained model across key metrics such as Acc $D_r$ (accuracy on the retain set), Acc $D_f$ (accuracy on the forget set), Acc $D_{test}$ (accuracy on the test set), and MIA score.

Our method achieves an average gap of 1.07, effectively unlearning the required knowledge while causing minimal degradation to the overall model performance. In comparison, SalUn, which utilizes all data splits for unlearning, achieves a slightly higher average gap of 1.15. Notably, our approach, relying solely on the forget set, outperforms SalUn, demonstrating superior efficiency in unlearning with-

*Table 2.* **Unlearning Performance for 10% Random Data Forgetting on CIFAR-10 using ResNet-18.** The results are reported as a±b, representing the mean (a) and standard deviation (b) across three independent trials. The Avg. Gap is the average performance difference between each unlearning model and the *Retrain model*, measured across Acc $D_r$, Acc $D_f$, Acc $D_{test}$, and MIA. ($\simeq$) denotes that smaller performance differences from the *Retrain model* are preferred. $^*$ indicates that the numbers are borrowed from (Fan et al., 2023). $^\dagger$ denotes the best results achieved through hyperparameter search. NegMerge achieves the lowest Avg. Gap, indicating performance closest to the *Retrain model*, which serves as the ideal benchmark.

| Method | Used Splits | Acc $D_r(\simeq)$ | Acc $D_f(\simeq)$ | Acc $D_{test}(\simeq)$ | MIA($\simeq$) | Avg. Gap($\downarrow$) |
|---|---|---|---|---|---|---|
| **Retrain** * | retrain | $100.00_{\pm0.00}$ | $94.76_{\pm0.69}$ | $94.26_{\pm0.02}$ | $12.88_{\pm0.09}$ | 0.00 |
| Random Labeling * | | $99.67_{\pm0.14}$ | $92.39_{\pm0.31}$ | $92.83_{\pm0.38}$ | $37.36_{\pm0.06}$ | 7.15 |
| Influence * | all | $99.20_{\pm0.22}$ | $98.93_{\pm0.28}$ | $93.20_{\pm1.03}$ | $2.67_{\pm0.01}$ | 4.06 |
| SalUn * | | $99.62_{\pm0.12}$ | $97.15_{\pm0.43}$ | $93.93_{\pm0.29}$ | $14.39_{\pm0.82}$ | 1.15 |
| Finetune * | | $99.88_{\pm0.08}$ | $99.37_{\pm0.55}$ | $94.06_{\pm0.27}$ | $2.70_{\pm0.01}$ | 3.78 |
| $\ell_1$-sparse * | retrain | $97.74_{\pm0.33}$ | $95.81_{\pm0.62}$ | $91.59_{\pm0.57}$ | $9.84_{\pm0.00}$ | 2.26 |
| Gradient Ascent * | | $99.50_{\pm0.38}$ | $99.31_{\pm0.54}$ | $94.01_{\pm0.47}$ | $1.70_{\pm0.01}$ | 4.12 |
| Boundary Shrink * | | $98.29_{\pm2.50}$ | $98.22_{\pm2.52}$ | $92.69_{\pm2.99}$ | $8.96_{\pm0.13}$ | 2.67 |
| Boundary Expanding * | forget | $99.42_{\pm0.33}$ | $99.41_{\pm0.30}$ | $93.85_{\pm1.02}$ | $7.47_{\pm1.15}$ | 2.76 |
| Random Labeling | | $99.99_{\pm0.00}$ | $99.98_{\pm0.02}$ | $95.04_{\pm0.11}$ | $2.15_{\pm1.94}$ | 4.19 |
| SalUn | | $99.88_{\pm0.04}$ | $99.89_{\pm0.04}$ | $94.42_{\pm0.05}$ | $9.51_{\pm2.07}$ | 2.20 |
| Task Arithmetic | | | | | | |
|   Single Best Model$^\dagger$ | | $98.36_{\pm0.51}$ | $94.85_{\pm0.16}$ | $91.49_{\pm0.80}$ | $10.91_{\pm0.72}$ | 1.62 |
|   Uniform Merge | forget | $98.70_{\pm0.91}$ | $95.83_{\pm2.17}$ | $92.36_{\pm1.16}$ | $10.14_{\pm2.93}$ | 1.75 |
|   TIES-Merging | | $98.38_{\pm0.17}$ | $95.45_{\pm0.32}$ | $92.23_{\pm0.14}$ | $9.36_{\pm0.31}$ | 1.96 |
|   MagMax | | $98.38_{\pm0.12}$ | $97.97_{\pm0.77}$ | $91.53_{\pm0.00}$ | $8.45_{\pm2.60}$ | 3.00 |
|   **NegMerge (ours)** | | $99.15_{\pm0.24}$ | $96.63_{\pm0.59}$ | $92.71_{\pm0.39}$ | $12.87_{\pm1.29}$ | **1.07** |

out depending on the retain set. In contrast, Task Arithmetic and merging methods, including Uniform Merge, TIES-Merging, and MagMax, exhibit larger gaps of 1.62, 1.75, 1.96, and 3.00, respectively. These results highlight that our method achieves a better balance between unlearning and preserving knowledge in the retain set. Furthermore, our approach ensures strong privacy protection, achieving an MIA score of 12.87, nearly identical to that of the retrained model. This demonstrates that the model effectively forgets targeted data without introducing privacy vulnerabilities.

Additional experiments are provided in Appendix B.3. We evaluate unlearning methods on CUB (Wah et al., 2011) and Tiny ImageNet (Le & Yang, 2015), demonstrating the effectiveness of NegMerge under diverse datasets (Table B8, Table B9). We also assess generalizability across backbone architectures by evaluating unlearning methods on VGG-16 (Simonyan, 2014) and Swin-T (Liu et al., 2021), where NegMerge consistently performs well (Table B10, Table B11). Finally, NegMerge maintains its advantage in different unlearning scenarios, including 50% random forgetting and class-wise forgetting, confirming its scalability and robustness (Table B12, Table B13).

### 4.3. Ablation Studies

**Effect of Sign Conflict on Unlearning Performance.** We argue that elements with consistent signs across multiple task vectors correspond to knowledge related to the forget

*Table 3.* **Impact of Sign Consensus Across Task Vectors.** The results present unlearning performance across multiple datasets, comparing three different methods. "*All*", Uniform Merge, uses all indices without regard to sign conflict, "*Conflict*" uses only indices with conflicting signs, and "*Consensus*", our proposed method, uses only indices with consistent signs across task vectors. The *Conflict* method degrades unlearning performance, while *Consensus* performs best, confirming the effectiveness of using only sign-consistent elements.

| Method | Cars | | DTD | | SUN397 | |
|---|---|---|---|---|---|---|
| | $D_f(\downarrow)$ | $D_r$ | $D_f(\downarrow)$ | $D_r$ | $D_f(\downarrow)$ | $D_r$ |
| All | 31.7 | 60.4 | 29.6 | 60.6 | 51.4 | 60.5 |
| Conflict | 40.2 | 60.2 | 31.9 | 60.3 | 58.3 | 60.9 |
| **Consensus** | 27.4 | 60.4 | 27.2 | 60.5 | 47.2 | 60.6 |

set, while elements with conflicting signs are less relevant to the forget set. To verify this, we compare unlearning performance when our method is applied in reverse. The experimental results are shown in Table 3. We use the CLIP ViT-B/32 model and the standard Task Arithmetic (Ilharco et al., 2022a). The *All* method refers to the Uniform Merge approach, which uses all elements without considering sign consensus. The *Conflict* method uses only elements with conflicting signs, while our proposed *Consensus* method uses only elements with consistent signs. The results show that the *Conflict* method significantly degrades unlearning performance, while the *All* method performs better than *Con-*

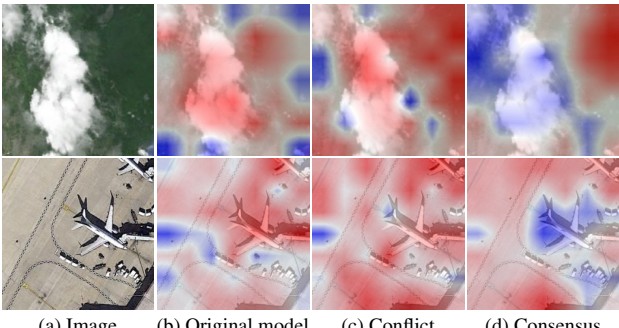

| (a) Image | (b) Original model | (c) Conflict | (d) Consensus |

*Figure 3.* **Visualization of the Impact of Sign Consensus.** Grad-CAM visualizations on the RESISC45 dataset compare the *Conflict*, *Consensus*, and original models. The first and second rows correspond to the *cloud* and *airplane* classes, respectively. Red areas indicate strong class relevance, while blue areas indicate weaker relevance. The *Consensus* model exhibits lower activation in class-relevant regions, indicating more effective unlearning.

*flict* but is outperformed by our *Consensus* method. These experimental results indicate that the design choice of merging only sign-consistent elements is effective. Full results are provided in Table B6.

In Figure 3, we demonstrate the effectiveness of our method using Grad-CAM (Selvaraju et al., 2017) visualizations on the RESISC45 dataset. We compare the *Conflict* and our *Consensus* methods, and include visualizations of the original model as a baseline. The red areas represent regions where the model strongly associates with the class label, while the blue areas indicate regions with less relevance. In the first row (*cloud* class), we observe that the *Conflict* method directs the model's attention to the cloud's location, resembling the behavior of the original model. In contrast, our method does not highlight the cloud's area, which suggests that the model has successfully forgotten its knowledge of the cloud. The same pattern appears in the second row for the *airplane* class. These visual results demonstrate that our proposed method is more effective for machine unlearning.

**Ratio of Zeroed Elements in the Merged Vector.** The proposed method masks out all elements that do not exhibit consistent signs across multiple task vectors, leaving only those with consistent signs. In Table 4, we investigate the proportion of zeroed-out elements in the final task vector $\tau$, where "zeroed-out" includes both elements that were already zero (*e.g.*, due to freezing during fine-tuning) and those masked during merging. Our observations show that as the number of merged models increases, the proportion of masked elements also increases, meaning that the sparsity of the final task vector $\tau$ increases with the inclusion of more models. Ultimately, the task vectors generated by our method modify only 5–10% of the weight elements in the

*Table 4.* **Sparsity and Unlearning Performance.** The results report the sparsity (*i.e.*, the percentage of zeroed elements in the final task vector $\tau$, denoted as %), Acc $D_f$, and Acc $D_r$, averaged over three runs using ViT-B/32. # indicates the number of task vectors used for merging. As more task vectors are merged, sparsity increases and Acc $D_f$ decreases, indicating improved unlearning performance.

| # | Cars | | | DTD | | | SUN397 | | |
|---|---|---|---|---|---|---|---|---|---|
| | % | $D_f(\downarrow)$ | $D_r$ | % | $D_f(\downarrow)$ | $D_r$ | % | $D_f(\downarrow)$ | $D_r$ |
| 30 | 90.3 | 27.4 | 60.4 | 92.9 | 27.2 | 60.5 | 92.2 | 47.2 | 60.6 |
| 25 | 89.7 | 26.6 | 60.2 | 92.2 | 27.3 | 60.5 | 91.7 | 47.0 | 60.4 |
| 20 | 88.8 | 26.1 | 60.1 | 91.5 | 27.1 | 60.4 | 90.9 | 47.8 | 60.6 |
| 15 | 87.5 | 26.0 | 60.0 | 90.2 | 27.7 | 60.4 | 89.6 | 47.7 | 60.5 |
| 10 | 84.9 | 26.6 | 59.9 | 87.5 | 27.8 | 60.4 | 87.2 | 48.7 | 60.6 |
| 5 | 77.1 | 30.5 | 60.4 | 79.9 | 28.8 | 60.5 | 81.5 | 49.7 | 60.5 |

*Table 5.* **Performance of Task Vector Addition.** The evaluated models are obtained by *adding* task vector $\tau$ to the original model. $^*$ denotes our reproduced results based on the configurations from (Ilharco et al., 2022a). $^\dagger$ represents the single model's best results achieved through hyperparameter tuning, including adjustments to data augmentation. $^\ddagger$ combines models in descending order of losses. NegMerge achieves high performance on both the forget and retain sets.

| Method | Cars | | DTD | | SUN397 | |
|---|---|---|---|---|---|---|
| | $D_f(\uparrow)$ | $D_r(\uparrow)$ | $D_f(\uparrow)$ | $D_r(\uparrow)$ | $D_f(\uparrow)$ | $D_r(\uparrow)$ |
| Pretrained | 59.6 | 66.7 | 43.9 | 66.7 | 63.3 | 66.7 |
| Task Arithmetic | | | | | | |
| *Paper config*$^*$ | 85.0 | 58.6 | 78.7 | 49.3 | 74.9 | 59.8 |
| Single Best$^\dagger$ | 86.6 | 52.7 | 76.9 | 48.4 | 76.5 | 55.7 |
| Uniform Merge | 87.2 | 55.3 | 79.0 | 52.8 | 76.0 | 57.1 |
| Greedy Merge$^\ddagger$ | 87.5 | 55.2 | 79.3 | 52.8 | 76.2 | 57.1 |
| **NegMerge (ours)** | 87.1 | **61.7** | 76.3 | **63.0** | 76.3 | **63.4** |

original model via negation, and interestingly, as sparsity increases, unlearning performance also improves. This suggests that leveraging a larger number of task vectors enables more accurate identification of specific elements corresponding to the forget set, allowing us to minimize performance degradation on the retain set. The full results are provided in Table B7. It is noteworthy that other merging methods have significantly fewer zeroed-out elements, resulting in lower unlearning performance, as shown in Table C2.

**Results of Task Vector Addition.** To better understand the characteristics of task vectors derived from our method and comparative techniques, we add these task vectors to the original model. The experimental results, presented in Table 5, reveal several key observations.

Most existing methods achieve high performance on the forget set but suffer significant degradation on the retain set. In contrast, our method maintains high performance on the retain set, which we consider a key factor in effective

*Table 6.* **Sparsity by Layer Depth.** Transformer layers are grouped into three depth ranges, and the average sparsity (%) is reported for each range across 30 models trained on the Cars dataset. Shallower layers exhibit higher sparsity.

| Layer Group | ViT-B/32 | | ViT-L/14 | |
|---|---|---|---|---|
| | Range | % | Range | % |
| Shallow | 0 – 3 | 73.4 | 0 – 7 | 83.5 |
| Middle | 4 – 7 | 68.4 | 8 – 15 | 76.4 |
| Deep | 8 – 11 | 62.9 | 16 – 23 | 73.3 |

machine unlearning. Specifically, for effective machine unlearning, the task vector must adjust the knowledge of the forget set while minimizing its impact on the retain set. Our experimental results demonstrate that our method effectively meets this requirement.

This characteristic of our approach appears to be closely related to the high sparsity of the task vector. As shown in Table C2, while other techniques modify 50–60% of the parameters, our method adjusts only 5–10%, enabling precise modification of performance on the forget set while minimizing the impact on the retain set. This finding is expected to aid future research in machine unlearning, and we plan to explore it further.

**NegMerge with Different Model Pools.** To evaluate the robustness of the proposed method to the composition of the model pool, we observe its performance using model pools constructed with various training configurations. Specifically, we conduct experiments by creating model pools with hyperparameter search spaces different from the original experimental setting. The results demonstrate that the proposed technique consistently outperforms the baseline, regardless of the model pool composition method. This indicates that the proposed method is robust to the composition of the model pool. The full experimental results are provided in Appendix D.

**Different Merging Operations.** Our method computes the final task vector by averaging the task vectors, as shown in Equation (1). The experimental results of using `max` or `min` operations instead of averaging are presented in Table C1. The results indicate that both `max` and `min` operations also improve unlearning performance compared to the baseline. However, our `avg` operation achieves the best performance. We hypothesize that averaging smooths out potential outliers in individual models during merging, resulting in a more stable and effective task vector.

**Ratio of Zeroed Elements by Layer Depth.** As shown in Table 6, sparsity, defined as the ratio of zeroed elements, decreases with layer depth in both ViT-B/32 and ViT-L/14. Shallower layers show higher sparsity, suggesting that they are more sensitive to hyperparameter variations and thus experience greater conflict across task vectors. We hypothe-

size that this is because general knowledge, parameterized within shallow layers, is more susceptible to distortion during fine-tuning. In contrast, deeper layers with task-specific knowledge are more robust to such variations (Morcos et al., 2018; Kornblith et al., 2019). These findings motivate future research on layer-wise unlearning strategies.

We further examine the sparsity across different components in Table C3. We also discuss the broader applicability of our method to LLMs and VLMs in Appendix F.

# 5. Conclusion and Limitation

We propose `NegMerge`, a novel machine unlearning technique that merges and leverages all fine-tuned models generated during validation, rather than discarding all but one. By introducing a sign consensus approach to identify and isolate parameters associated with the forget set, `NegMerge` modifies only the parameters strongly related to the forget set, achieving effective unlearning while preserving the knowledge of the retain set. `NegMerge` achieves state-of-the-art performance across 12 datasets and 4 architectures. Additionally, by improving the validation process of existing techniques, `NegMerge` effectively leverages multiple models without requiring additional computational resources. Although empirically validated, `NegMerge` currently lacks theoretical justification. Future work will focus on theoretical validation and analytical insights for method extension.

# Acknowledgments

Most experiments were conducted on the NAVER Smart Machine Learning (NSML) platform (Sung et al., 2017). This work was supported by the National Research Foundation of Korea (NRF) grant funded by the Korean government (MSIT) (RS-2024-00350430). We thank the researchers at NAVER AI Lab, particularly Taekyung Kim, Changdae Oh, and Yong-Hyun Park, for their invaluable feedback and the VRL Lab at Sogang University for their support.

# Impact Statement

This paper contributes to advancing the field of machine unlearning. Our method offers positive societal impacts in terms of privacy and data management. Notably, unlike existing approaches, it utilizes all the fine-tuned models generated during the validation process instead of discarding them, making it more resource-friendly. This could lead to various positive effects, such as reducing carbon emissions. However, if malicious users gain direct access to the model, there is a risk of adverse effects, such as deleting knowledge that should not be erased. This risk can be mitigated by implementing additional security protocols to ensure that only authorized individuals have access to the model.

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

# Appendix

This appendix includes the following materials: 1) Implementation Details (Appendix A), 2) Further Experimental Results (Appendix B), 3) Additional Ablation Studies (Appendix C), 4) Results in Diverse Model Pool (Appendix D), 5) Theoretical Analysis (Appendix E), 6) Applicability Beyond Image Classification (Appendix F).

## A. Implementation Details

In the CLIP scenario, for fine-tuning, we set the batch size to 128 and use a learning rate of 1e-5 with a cosine annealing schedule. We utilize the AdamW optimizer, applying a weight decay of 0.1. Following Task Arithmetic (Ilharco et al., 2022a), we freeze the final classification layer of CLIP's text encoder during fine-tuning. With the observation that freezing the classification layer does not affect accuracy (Ilharco et al., 2022b), we do not consider unfreezing the final layer of CLIP's text encoder. Attention layers are also frozen (Ye et al., 2023; Tang et al., 2024). We fine-tune the models by adjusting the configurations of RandAugment. Specifically, we vary the number of sequential augmentation transformations (ranging from 1 to 3) and the magnitude of these transformations (ranging from 1 to 10). A total of 30 models are fine-tuned. Unlike previous works, we incorporate data augmentation directly into the fine-tuning process, which requires adjusting the number of training epochs to better accommodate the augmented data. Consequently, the number of training epochs is set as follows: 70 epochs for Cars, 100 epochs for DTD, 40 epochs for EuroSAT, GTSRB, RESISC45, SUN397, and 30 epochs for MNIST and SVHN.

In the standard image classifier unlearning scenario, we fine-tune models with varied hyperparameters. For CIFAR-10, we use ResNet-18 with a batch size of 256 and a learning rate of 0.05, and VGG-16 with a batch size of 64 and a learning rate of 0.01. Instead of data augmentation, we adjust training settings, setting the number of epochs to 40, 50, or 60, weight decay to 1e-4, 5e-5, and 1e-5, and label smoothing to 0, 0.05, or 0.1, resulting in 27 fine-tuned models. Similarly, for CUB with ResNet-18, we use a batch size of 64, a learning rate of 0.01, and vary epochs to 5, 10, or 20, weight decay to 1e-4, 5e-5, and 1e-5, and label smoothing to 0, 0.05, and 0.1, producing another 27 models.

## B. Further Experimental Results

### B.1. Full Results

Table B1, Table B2, and Table B3 show the full accuracy results for the eight datasets (Cars, DTD, EuroSAT, GTSRB, MNIST, RESISC45, SUN397, and SVHN) and the three CLIP models we examine. Similarly, Table B4 and Table B5 provide accuracy results for Linear Task Arithmetic (Ortiz-Jimenez et al., 2024) on these datasets for two CLIP models.

*Table B1.* **ViT-B/32 Task Arithmetic Results.**

| Method | Cars | | DTD | | EuroSAT | | GTSRB | | MNIST | | RESISC45 | | SUN397 | | SVHN | |
|---|---|---|---|---|---|---|---|---|---|---|---|---|---|---|---|---|
| | $D_f(\downarrow)$ | $D_r$ | $D_f(\downarrow)$ | $D_r$ | $D_f(\downarrow)$ | $D_r$ | $D_f(\downarrow)$ | $D_r$ | $D_f(\downarrow)$ | $D_r$ | $D_f(\downarrow)$ | $D_r$ | $D_f(\downarrow)$ | $D_r$ | $D_f(\downarrow)$ | $D_r$ |
| Task Arithmetic[†] | 29.0 | 59.9 | 30.4 | 60.8 | 10.4 | 60.9 | 9.1 | 60.9 | 21.2 | 60.6 | 30.7 | 60.8 | 50.6 | 59.9 | 7.6 | 60.9 |
| Uniform Merge | 31.7 | 60.4 | 29.6 | 60.6 | 8.9 | 60.8 | 7.0 | 60.0 | 20.5 | 61.4 | 23.8 | 60.1 | 51.4 | 60.5 | 7.3 | 60.7 |
| Greedy Merge[‡] | 31.0 | 60.3 | 29.5 | 60.6 | 9.4 | 60.8 | 8.4 | 60.5 | 21.3 | 62.0 | 28.3 | 60.7 | 51.4 | 60.4 | 7.2 | 60.7 |
| TIES-Merging | 34.0 | 60.3 | 33.1 | 61.3 | 11.6 | 61.1 | 10.2 | 61.3 | 26.1 | 62.4 | 33.4 | 61.0 | 53.8 | 60.3 | 7.5 | 60.9 |
| MagMax | 35.6 | 60.6 | 31.9 | 61.1 | 10.5 | 60.7 | 8.4 | 60.8 | 20.1 | 60.7 | 30.7 | 60.6 | 55.4 | 61.1 | 9.3 | 62.0 |
| **NegMerge (ours)** | 27.4 | 60.4 | 27.2 | 60.5 | 7.9 | 60.2 | 6.2 | 60.0 | 20.5 | 59.9 | 22.6 | 60.5 | 47.2 | 60.6 | 7.2 | 60.9 |

*Table B2.* **ViT-B/16 Task Arithmetic Results.**

| Method | Cars | | DTD | | EuroSAT | | GTSRB | | MNIST | | RESISC45 | | SUN397 | | SVHN | |
|---|---|---|---|---|---|---|---|---|---|---|---|---|---|---|---|---|
| | $D_f(\downarrow)$ | $D_r$ | $D_f(\downarrow)$ | $D_r$ | $D_f(\downarrow)$ | $D_r$ | $D_f(\downarrow)$ | $D_r$ | $D_f(\downarrow)$ | $D_r$ | $D_f(\downarrow)$ | $D_r$ | $D_f(\downarrow)$ | $D_r$ | $D_f(\downarrow)$ | $D_r$ |
| Task Arithmetic[†] | 31.6 | 63.8 | 26.1 | 63.8 | 7.6 | 64.3 | 7.7 | 64.5 | 8.9 | 64.0 | 27.2 | 64.4 | 49.1 | 63.7 | 6.9 | 63.9 |
| Uniform Merge | 32.9 | 64.6 | 26.3 | 64.5 | 9.8 | 64.8 | 7.0 | 64.1 | 13.9 | 65.0 | 25.6 | 64.7 | 49.7 | 64.6 | 6.9 | 64.7 |
| Greedy Merge[‡] | 32.9 | 64.6 | 25.0 | 63.7 | 9.9 | 64.7 | 7.0 | 64.1 | 12.4 | 64.8 | 25.6 | 64.6 | 51.1 | 65.1 | 6.9 | 64.7 |
| TIES-Merging | 39.4 | 65.0 | 27.4 | 64.0 | 10.2 | 64.8 | 8.6 | 64.6 | 11.1 | 64.9 | 33.6 | 65.3 | 53.2 | 64.8 | 6.7 | 64.3 |
| MagMax | 38.4 | 64.8 | 26.6 | 63.9 | 10.2 | 65.0 | 9.0 | 64.9 | 14.6 | 64.4 | 36.6 | 66.0 | 53.5 | 65.0 | 6.7 | 64.3 |
| **NegMerge (ours)** | 28.8 | 64.8 | 25.2 | 64.5 | 9.8 | 65.9 | 7.1 | 64.4 | 10.7 | 63.8 | 20.3 | 63.9 | 45.2 | 64.4 | 7.0 | 64.6 |

*Table B3.* **ViT-L/14 Task Arithmetic Results.**

| Method | Cars | | DTD | | EuroSAT | | GTSRB | | MNIST | | RESISC45 | | SUN397 | | SVHN | |
|---|---|---|---|---|---|---|---|---|---|---|---|---|---|---|---|---|
| | $D_f(\downarrow)$ | $D_r$ | $D_f(\downarrow)$ | $D_r$ | $D_f(\downarrow)$ | $D_r$ | $D_f(\downarrow)$ | $D_r$ | $D_f(\downarrow)$ | $D_r$ | $D_f(\downarrow)$ | $D_r$ | $D_f(\downarrow)$ | $D_r$ | $D_f(\downarrow)$ | $D_r$ |
| Task Arithmetic[†] | 34.6 | 72.2 | 24.7 | 71.3 | 5.4 | 72.5 | 3.0 | 71.6 | 10.3 | 73.6 | 17.0 | 71.7 | 51.6 | 71.9 | 6.7 | 71.9 |
| Uniform Merge | 29.1 | 71.8 | 23.5 | 71.4 | 8.2 | 72.1 | 3.1 | 71.5 | 9.9 | 72.4 | 13.9 | 71.5 | 50.5 | 72.3 | 6.7 | 72.2 |
| Greedy Merge[‡] | 28.2 | 71.5 | 23.9 | 71.5 | 7.3 | 73.0 | 3.1 | 71.7 | 9.9 | 72.8 | 11.5 | 71.0 | 51.1 | 72.3 | 6.8 | 72.1 |
| TIES-Merging | 48.2 | 73.1 | 25.5 | 71.5 | 9.2 | 72.4 | 4.1 | 72.6 | 10.3 | 73.0 | 21.0 | 72.0 | 56.6 | 72.8 | 6.8 | 71.9 |
| MagMax | 39.2 | 72.0 | 28.7 | 72.7 | 9.9 | 73.6 | 4.2 | 72.5 | 10.7 | 73.5 | 20.6 | 72.2 | 53.7 | 72.1 | 6.7 | 71.9 |
| **NegMerge (ours)** | 32.7 | 71.9 | 23.9 | 71.9 | 9.1 | 72.1 | 2.8 | 71.3 | 10.9 | 73.6 | 8.8 | 70.9 | 43.6 | 72.1 | 6.8 | 72.8 |

*Table B4.* **ViT-B/32 Linear Task Arithmetic Results.**

| Method | Cars | | DTD | | EuroSAT | | GTSRB | | MNIST | | RESISC45 | | SUN397 | | SVHN | |
|---|---|---|---|---|---|---|---|---|---|---|---|---|---|---|---|---|
| | $D_f(\downarrow)$ | $D_r$ | $D_f(\downarrow)$ | $D_r$ | $D_f(\downarrow)$ | $D_r$ | $D_f(\downarrow)$ | $D_r$ | $D_f(\downarrow)$ | $D_r$ | $D_f(\downarrow)$ | $D_r$ | $D_f(\downarrow)$ | $D_r$ | $D_f(\downarrow)$ | $D_r$ |
| Task Arithmetic[†] | 13.5 | 60.2 | 15.2 | 59.7 | 0.1 | 60.3 | 0.2 | 60.2 | 0.1 | 61.0 | 2.6 | 59.6 | 38.8 | 59.9 | 0.7 | 60.5 |
| Uniform Merge | 14.2 | 60.4 | 15.3 | 60.2 | 0.0 | 60.3 | 0.2 | 60.8 | 0.0 | 60.2 | 2.6 | 60.2 | 39.7 | 60.4 | 0.8 | 61.3 |
| Greedy Merge[‡] | 14.4 | 60.3 | 15.8 | 60.2 | 0.0 | 60.4 | 0.2 | 60.2 | 0.0 | 60.5 | 2.6 | 60.3 | 36.3 | 59.7 | 0.7 | 60.6 |
| TIES-Merging | 19.3 | 60.4 | 16.5 | 60.0 | 0.3 | 60.5 | 0.2 | 60.5 | 0.0 | 60.4 | 5.6 | 60.4 | 42.8 | 60.3 | 0.8 | 60.5 |
| MagMax | 22.6 | 61.0 | 16.5 | 60.1 | 0.2 | 60.6 | 0.2 | 60.8 | 0.1 | 61.5 | 4.2 | 60.1 | 46.1 | 60.9 | 0.7 | 60.4 |
| **NegMerge (ours)** | 12.1 | 60.6 | 15.6 | 60.4 | 0.0 | 60.9 | 0.2 | 61.2 | 0.0 | 60.9 | 1.6 | 60.1 | 34.0 | 59.8 | 0.7 | 60.8 |

*Table B5.* **ViT-B/16 Linear Task Arithmetic Results.**

| Method | Cars | | DTD | | EuroSAT | | GTSRB | | MNIST | | RESISC45 | | SUN397 | | SVHN | |
|---|---|---|---|---|---|---|---|---|---|---|---|---|---|---|---|---|
| | $D_f(\downarrow)$ | $D_r$ | $D_f(\downarrow)$ | $D_r$ | $D_f(\downarrow)$ | $D_r$ | $D_f(\downarrow)$ | $D_r$ | $D_f(\downarrow)$ | $D_r$ | $D_f(\downarrow)$ | $D_r$ | $D_f(\downarrow)$ | $D_r$ | $D_f(\downarrow)$ | $D_r$ |
| Task Arithmetic[†] | 5.3 | 64.4 | 10.2 | 63.8 | 0.0 | 64.8 | 0.0 | 64.5 | 0.1 | 67.0 | 2.0 | 63.6 | 37.4 | 64.7 | 0.5 | 64.1 |
| Uniform Merge | 5.0 | 64.7 | 10.1 | 64.2 | 0.1 | 66.0 | 0.0 | 66.0 | 0.1 | 67.4 | 1.6 | 64.0 | 37.5 | 65.0 | 0.4 | 64.8 |
| Greedy Merge[‡] | 5.0 | 64.8 | 10.3 | 64.1 | 0.0 | 64.5 | 0.0 | 64.5 | 0.1 | 66.8 | 1.5 | 63.9 | 37.1 | 65.0 | 0.4 | 64.2 |
| TIES-Merging | 7.4 | 64.3 | 11.7 | 64.6 | 0.1 | 65.3 | 0.0 | 65.4 | 0.1 | 66.9 | 4.1 | 64.4 | 43.8 | 65.7 | 0.4 | 64.3 |
| MagMax | 8.8 | 64.6 | 12.3 | 64.9 | 0.0 | 64.9 | 0.0 | 64.9 | 0.1 | 67.2 | 5.1 | 64.9 | 42.5 | 65.4 | 0.4 | 64.6 |
| **NegMerge (ours)** | 6.6 | 65.9 | 10.4 | 64.5 | 0.0 | 65.9 | 0.0 | 66.0 | 0.1 | 66.9 | 1.1 | 64.5 | 34.1 | 64.7 | 0.5 | 64.8 |

Table B6 extends Table 3 by showing results for all eight datasets, analyzing the impact of sign conflicts in weights during unlearning. Table B7 expands on Table 4, providing zero ratios, forget set accuracy (Acc $D_f$), and retain set accuracy (Acc $D_r$) across different numbers of task vectors used for merging.

*Table B6.* **Impact of Sign Conflict in Weights for Unlearning.** This table presents unlearning performance across various datasets using CLIP ViT-B/32, comparing three different methods. "All," Uniform Merge, uses all indices regardless of sign conflict, "Conflict" uses only indices with conflicting signs, and "Non-conflict," our proposed method, uses only indices with consistent signs across task vectors.

| Method | Cars | | DTD | | EuroSAT | | GTSRB | | MNIST | | RESISC45 | | SUN397 | | SVHN | |
|---|---|---|---|---|---|---|---|---|---|---|---|---|---|---|---|---|
| | $D_f(\downarrow)$ | $D_r$ | $D_f(\downarrow)$ | $D_r$ | $D_f(\downarrow)$ | $D_r$ | $D_f(\downarrow)$ | $D_r$ | $D_f(\downarrow)$ | $D_r$ | $D_f(\downarrow)$ | $D_r$ | $D_f(\downarrow)$ | $D_r$ | $D_f(\downarrow)$ | $D_r$ |
| All | 31.7 | 60.4 | 29.6 | 60.6 | 8.9 | 60.8 | 7.0 | 60.0 | 20.5 | 61.4 | 23.8 | 60.1 | 51.4 | 60.5 | 7.3 | 60.7 |
| Conflict | 40.2 | 60.2 | 31.9 | 60.3 | 11.1 | 60.7 | 9.1 | 60.6 | 24.0 | 61.9 | 32.3 | 60.2 | 58.3 | 60.9 | 8.8 | 60.6 |
| **Non-conflict** | 27.4 | 60.4 | 27.2 | 60.5 | 7.9 | 60.2 | 6.2 | 60.0 | 20.5 | 59.9 | 22.6 | 60.5 | 47.2 | 60.6 | 7.2 | 60.9 |

## B.2. Full Charts of CLIP Unlearning Scenario

We provide the complete trade-off graphs illustrating the forget set's accuracy (*i.e.*, 1 - accuracy) versus the retain set's accuracy (Figure B1), extending the partial illustration presented in the main paper (Figure 1). Each graph denotes the trade-offs for different datasets, including Cars, DTD, EuroSAT, GTSRB, MNIST, RESISC45, SUN397, and SVHN. Our method enjoys the best trade-offs among competing methods across most of the datasets.

*Table B7.* **Ratio of Zeroed Elements based on the Number of Merged Models.** The results, averaged over three runs with standard deviations (std, $\pm$), were obtained using ViT-B/32 Task Arithmetic.

(a) Part 1: Cars, DTD, EuroSAT, GTSRB

| # | Cars | | | DTD | | | EuroSAT | | | GTSRB | | |
|---|---|---|---|---|---|---|---|---|---|---|---|---|
| | % | Acc $D_f(\downarrow)$ | Acc $D_r$ | % | Acc $D_f(\downarrow)$ | Acc $D_r$ | % | Acc $D_f(\downarrow)$ | Acc $D_r$ | % | Acc $D_f(\downarrow)$ | Acc $D_r$ |
| 30 | 90.3 | 27.4 | 60.4 | 92.9 | 27.2 | 60.5 | 94.1 | 7.9 | 60.2 | 94.8 | 6.2 | 60.0 |
| 25 | $89.7_{\pm0.04}$ | $26.6_{\pm0.06}$ | $60.2_{\pm0.02}$ | $92.2_{\pm0.10}$ | $27.2_{\pm0.28}$ | $60.5_{\pm0.10}$ | $93.4_{\pm0.09}$ | $8.3_{\pm0.06}$ | $60.3_{\pm0.05}$ | $94.1_{\pm0.02}$ | $6.5_{\pm0.01}$ | $60.1_{\pm0.03}$ |
| 20 | $88.8_{\pm0.21}$ | $26.1_{\pm0.66}$ | $60.1_{\pm0.09}$ | $91.5_{\pm0.13}$ | $27.1_{\pm0.13}$ | $60.4_{\pm0.04}$ | $92.7_{\pm0.05}$ | $8.8_{\pm0.06}$ | $60.6_{\pm0.03}$ | $93.3_{\pm0.03}$ | $6.7_{\pm0.06}$ | $60.3_{\pm0.01}$ |
| 15 | $87.5_{\pm0.17}$ | $26.0_{\pm0.20}$ | $60.0_{\pm0.07}$ | $90.2_{\pm0.12}$ | $27.7_{\pm0.09}$ | $60.4_{\pm0.03}$ | $90.8_{\pm0.28}$ | $8.3_{\pm0.11}$ | $60.4_{\pm0.07}$ | $91.9_{\pm0.09}$ | $6.9_{\pm0.17}$ | $60.3_{\pm0.01}$ |
| 10 | $84.9_{\pm0.51}$ | $26.6_{\pm0.82}$ | $59.9_{\pm0.14}$ | $87.5_{\pm0.31}$ | $27.8_{\pm0.09}$ | $60.4_{\pm0.12}$ | $88.3_{\pm0.51}$ | $8.9_{\pm0.21}$ | $60.6_{\pm0.11}$ | $89.2_{\pm0.19}$ | $8.1_{\pm0.08}$ | $60.6_{\pm0.04}$ |
| 5 | $77.1_{\pm0.30}$ | $30.5_{\pm0.42}$ | $60.4_{\pm0.09}$ | $79.9_{\pm0.60}$ | $28.8_{\pm0.39}$ | $60.5_{\pm0.14}$ | $81.5_{\pm0.26}$ | $9.3_{\pm0.55}$ | $60.6_{\pm0.10}$ | $81.9_{\pm0.32}$ | $8.3_{\pm0.35}$ | $60.5_{\pm0.08}$ |

(b) Part 2: MNIST, RESISC45, SUN397, SVHN

| # | MNIST | | | RESISC45 | | | SUN397 | | | SVHN | | |
|---|---|---|---|---|---|---|---|---|---|---|---|---|
| | % | Acc $D_f(\downarrow)$ | Acc $D_r$ | % | Acc $D_f(\downarrow)$ | Acc $D_r$ | % | Acc $D_f(\downarrow)$ | Acc $D_r$ | % | Acc $D_f(\downarrow)$ | Acc $D_r$ |
| 30 | 94.0 | 20.5 | 59.9 | 92.9 | 22.6 | 60.5 | 92.2 | 47.2 | 60.6 | 92.4 | 7.2 | 60.9 |
| 25 | $93.3_{\pm0.10}$ | $20.5_{\pm0.46}$ | $60.2_{\pm0.06}$ | $92.3_{\pm0.04}$ | $23.7_{\pm0.06}$ | $60.6_{\pm0.02}$ | $91.7_{\pm0.09}$ | $47.0_{\pm0.02}$ | $60.4_{\pm0.04}$ | $91.7_{\pm0.02}$ | $7.1_{\pm0.01}$ | $60.7_{\pm0.01}$ |
| 20 | $92.5_{\pm0.08}$ | $19.6_{\pm0.46}$ | $59.9_{\pm0.07}$ | $91.5_{\pm0.06}$ | $22.7_{\pm0.08}$ | $60.5_{\pm0.02}$ | $90.9_{\pm0.12}$ | $47.8_{\pm0.10}$ | $60.6_{\pm0.04}$ | $90.7_{\pm0.04}$ | $7.0_{\pm0.02}$ | $60.4_{\pm0.03}$ |
| 15 | $91.0_{\pm0.06}$ | $20.8_{\pm0.49}$ | $60.1_{\pm0.30}$ | $90.1_{\pm0.09}$ | $23.2_{\pm0.18}$ | $60.5_{\pm0.03}$ | $89.6_{\pm0.33}$ | $47.7_{\pm0.54}$ | $60.5_{\pm0.19}$ | $89.2_{\pm0.27}$ | $7.1_{\pm0.04}$ | $60.7_{\pm0.08}$ |
| 10 | $88.2_{\pm0.47}$ | $19.9_{\pm0.77}$ | $60.5_{\pm0.15}$ | $87.6_{\pm0.28}$ | $22.6_{\pm0.79}$ | $60.4_{\pm0.08}$ | $87.2_{\pm0.31}$ | $48.7_{\pm0.41}$ | $60.6_{\pm0.11}$ | $86.0_{\pm0.20}$ | $7.2_{\pm0.01}$ | $60.8_{\pm0.08}$ |
| 5 | $81.4_{\pm0.25}$ | $23.0_{\pm0.36}$ | $62.4_{\pm0.02}$ | $80.2_{\pm0.52}$ | $24.1_{\pm1.24}$ | $60.4_{\pm0.20}$ | $81.5_{\pm0.10}$ | $49.7_{\pm0.08}$ | $60.5_{\pm0.11}$ | $78.9_{\pm0.18}$ | $7.2_{\pm0.03}$ | $60.6_{\pm0.08}$ |

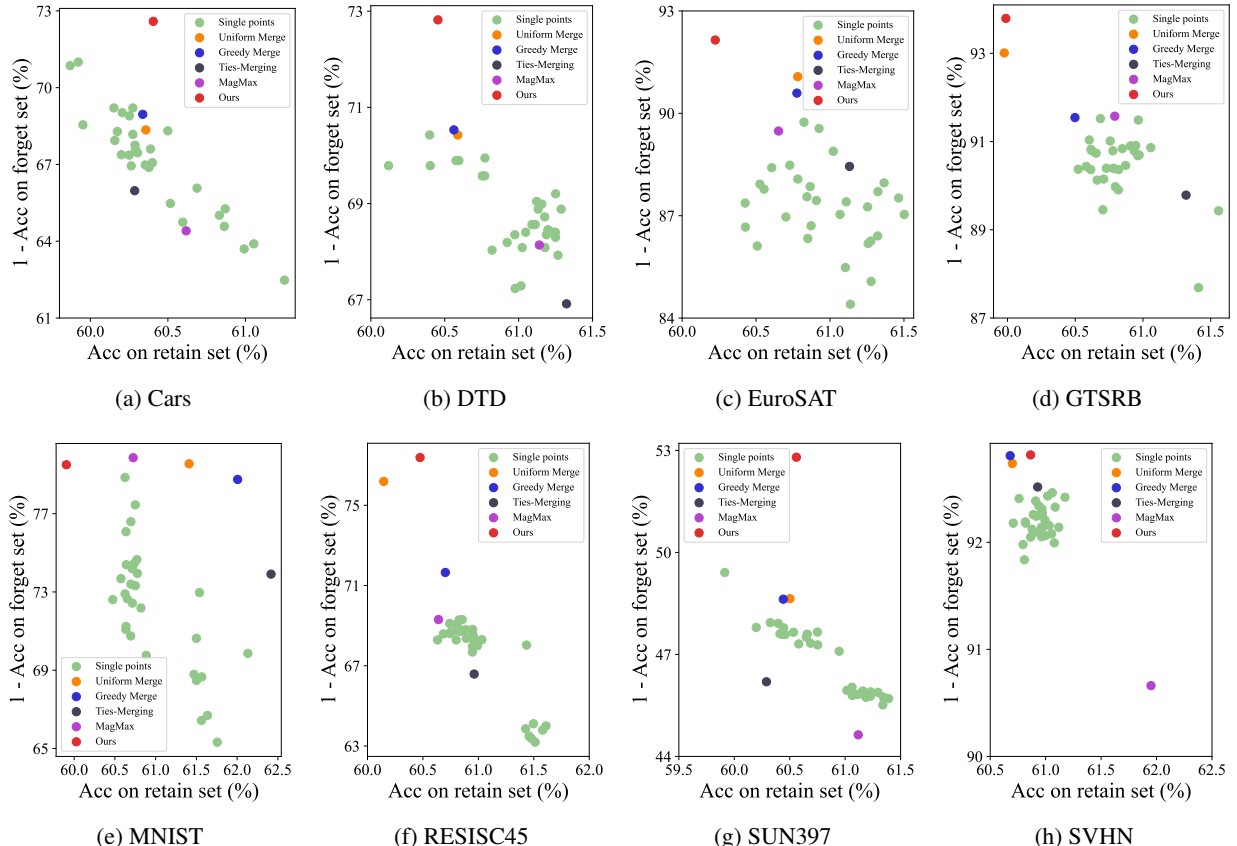

(a) Cars    (b) DTD    (c) EuroSAT    (d) GTSRB

(e) MNIST    (f) RESISC45    (g) SUN397    (h) SVHN

*Figure B1.* **Comparison of Merged Models on ViT-B/32.** Performance metrics for merged models showing accuracy on the retain set and forget set across different models. Methods positioned towards the upper right corner are generally considered to be better performers.

## B.3. Additional Standard Classifier Unlearning Scenario Results

**Performance on Different Datasets.** Table B8 compares unlearning methods on CUB (Wah et al., 2011) with ResNet-18 for 10% random data forgetting. This experiment is important as it validates the effectiveness of `NegMerge` in fine-grained image classification. Table B9 shows similar results on Tiny ImageNet (Le & Yang, 2015) using ResNet-18.

*Table B8.* **Unlearning Performance for 10% Random Data Forgetting on CUB using ResNet-18.**

| Method | Used Splits | Acc $D_r(\simeq)$ | Acc $D_f(\simeq)$ | Acc $D_{test}(\simeq)$ | MIA$(\simeq)$ | Avg. Gap$(\downarrow)$ |
|---|---|---|---|---|---|---|
| **Retrain** | retain | 78.55 | 56.43 | 74.61 | 80.47 | 0.00 |
| Gradient Ascent | | 66.75 | 57.26 | 66.60 | 67.61 | 8.38 |
| Boundary Shrink | | 66.88 | 61.60 | 64.14 | 100.00 | 11.71 |
| Boundary Expanding | forget | 65.32 | 61.60 | 58.80 | 73.62 | 10.27 |
| Random Labeling | | 64.13 | 57.43 | 59.54 | 71.79 | 9.79 |
| SalUn | | 66.69 | 59.60 | 63.88 | 74.46 | 7.94 |
| Task Arithmetic | | | | | | |
|   Single Best Model[†] | forget | 74.68 | 58.60 | 70.56 | 100.00 | 7.41 |
|   Uniform Merge | | 73.94 | 56.93 | 69.78 | 100.00 | 7.37 |
|   **NegMerge (ours)** | | 74.64 | 58.26 | 70.69 | 100.00 | **7.30** |

*Table B9.* **Unlearning Performance for 10% Random Data Forgetting on Tiny ImageNet using ResNet-18.**

| Method | Used Splits | Acc $D_r(\simeq)$ | Acc $D_f(\simeq)$ | Acc $D_{test}(\simeq)$ | MIA$(\simeq)$ | Avg. Gap$(\downarrow)$ |
|---|---|---|---|---|---|---|
| **Retrain \*** | retain | 99.98 | 63.60 | 63.67 | 63.77 | 0.00 |
| Random Labeling | forget | 76.43 | 76.09 | 58.09 | 32.33 | 18.27 |
| SalUn | | 73.61 | 73.81 | 56.68 | 30.59 | 19.19 |
| Task Arithmetic | | | | | | |
|   Single Best Model[†] | forget | 77.54 | 73.63 | 59.62 | 30.14 | 17.54 |
|   **NegMerge (ours)** | | 75.95 | 71.87 | 58.58 | 31.34 | **17.46** |

**Performance on Different Models.** Table B10 and Table B11 show results on CIFAR-10 using VGG-16 (Simonyan, 2014) and Swin-T (Liu et al., 2021), respectively. In both cases, `NegMerge` effectively unlearns the forget set $D_f$ using only the forget set itself. These results demonstrate that our method generalizes well across different model architectures.

*Table B10.* **Unlearning Performance for 10% Random Data Forgetting on CIFAR-10 using VGG-16.**

| Method | Used Splits | Acc $D_r(\simeq)$ | Acc $D_f(\simeq)$ | Acc $D_{test}(\simeq)$ | MIA$(\simeq)$ | Avg. Gap$(\downarrow)$ |
|---|---|---|---|---|---|---|
| **Retrain \*** | retain | 99.99 | 94.02 | 93.06 | 10.36 | 0.00 |
| Gradient Ascent \* | | 99.37 | 99.07 | 93.63 | 1.36 | 3.81 |
| Boundary Shrink \* | forget | 99.40 | 99.20 | 93.68 | 1.38 | 3.84 |
| Boundary Expanding \* | | 99.39 | 99.20 | 93.68 | 1.42 | 3.84 |
| Task Arithmetic | | | | | | |
|   Single Best Model[†] | forget | 97.26 | 94.90 | 90.10 | 10.34 | 1.64 |
|   **NegMerge (ours)** | | 98.00 | 95.74 | 91.01 | 10.10 | **1.50** |

*Table B11.* **Unlearning Performance for 10% Random Data Forgetting on CIFAR-10 using Swin-T.**

| Method | Used Splits | Acc $D_r(\simeq)$ | Acc $D_f(\simeq)$ | Acc $D_{test}(\simeq)$ | MIA$(\simeq)$ | Avg. Gap$(\downarrow)$ |
|---|---|---|---|---|---|---|
| **Retrain** | retain | 100.00 | 97.76 | 97.67 | 4.74 | 0.00 |
| Random Labeling | forget | 99.96 | 99.96 | 97.71 | 0.64 | 1.60 |
| SalUn | | 98.98 | 99.04 | 96.40 | 3.36 | 1.24 |
| Task Arithmetic | | | | | | |
|   Single Best Model[†] | forget | 98.50 | 97.84 | 95.98 | 4.04 | 0.99 |
|   **NegMerge (ours)** | | 98.79 | 97.80 | 95.93 | 4.60 | **0.78** |

**Performance on Different Scenarios.** Table B12 presents a comparison of various unlearning techniques for 50% random data forgetting on CIFAR-10 using ResNet-18. Table B13 compares various unlearning techniques for class-wise forgetting on CIFAR-10 using ResNet-18. These results highlight the scalability of `NegMerge` across diverse unlearning scenarios.

*Table B12.* **Unlearning Performance for 50% Random Data Forgetting on CIFAR-10 using ResNet-18.**

| Method | Used Splits | Acc $D_r(\simeq)$ | Acc $D_f(\simeq)$ | Acc $D_{test}(\simeq)$ | MIA($\simeq$) | Avg. Gap($\downarrow$) |
|---|---|---|---|---|---|---|
| **Retrain** | retain | 100.00 | 92.10 | 91.70 | 19.30 | 0.00 |
| Random Labeling | forget | 99.80 | 99.90 | 94.70 | 2.20 | 7.03 |
| SalUn | | 99.60 | 99.60 | 94.20 | 4.40 | 6.33 |
| Task Arithmetic | | | | | | |
| Single Best Model[†] | forget | 98.40 | 97.90 | 92.60 | 5.60 | 5.50 |
| **NegMerge (ours)** | | 96.80 | 96.50 | 91.50 | 6.30 | **5.20** |

*Table B13.* **Unlearning Performance for Class-wise Forgetting on CIFAR-10 using ResNet-18.**

| Method | Used Splits | Acc $D_r(\simeq)$ | Acc $D_f(\simeq)$ | Acc $D_{test}(\simeq)$ | MIA($\simeq$) | Avg. Gap($\downarrow$) |
|---|---|---|---|---|---|---|
| **Retrain** | retain | 100.00 | 0.00 | 92.50 | 100.00 | 0.00 |
| Random Labeling | forget | 83.00 | 10.10 | 70.90 | 99.50 | 12.30 |
| SalUn | | 86.50 | 10.80 | 74.10 | 100.00 | 10.68 |
| Task Arithmetic | | | | | | |
| Single Best Model[†] | forget | 95.30 | 0.10 | 80.60 | 100.00 | 4.18 |
| **NegMerge (ours)** | | 96.80 | 0.80 | 81.80 | 99.80 | **3.73** |

## C. Additional Ablation Studies

Table C1 compares ways to derive the improved final task vector $\tau_{\text{merged}}$. We found that our originally proposed averaging method performed the best. This is likely because averaging helps smooth out potential outliers in individual models during merging, resulting in a more stable and effective task vector.

*Table C1.* **Results When Different Merging Operations Are Used.** `NegMerge` (min), `NegMerge` (max), and `NegMerge` (avg) represent merging minimum, maximum, and average of task vectors elements, respectively. The experimental results are obtained using CLIP ViT-B/32.

| Method | Cars | | DTD | | EuroSAT | | GTSRB | | MNIST | | RESISC45 | | SUN397 | | SVHN | |
|---|---|---|---|---|---|---|---|---|---|---|---|---|---|---|---|---|
| | $D_f(\downarrow)$ | $D_r$ | $D_f(\downarrow)$ | $D_r$ | $D_f(\downarrow)$ | $D_r$ | $D_f(\downarrow)$ | $D_r$ | $D_f(\downarrow)$ | $D_r$ | $D_f(\downarrow)$ | $D_r$ | $D_f(\downarrow)$ | $D_r$ | $D_f(\downarrow)$ | $D_r$ |
| Task Arithmetic[†] | 29.0 | 59.9 | 30.4 | 60.8 | 10.4 | 60.9 | 9.1 | 60.9 | 21.2 | 60.6 | 30.7 | 60.8 | 50.6 | 59.9 | 7.6 | 60.9 |
| NegMerge (min) | 26.5 | 59.9 | 27.9 | 60.6 | 10.3 | 60.8 | 8.3 | 60.7 | 25.2 | 61.0 | 20.1 | 60.0 | 46.8 | 60.2 | 8.2 | 61.0 |
| NegMerge (max) | 28.2 | 60.3 | 27.4 | 60.4 | 10.7 | 60.9 | 7.5 | 60.4 | 28.7 | 60.2 | 26.0 | 61.0 | 48.3 | 60.7 | 7.3 | 60.8 |
| **NegMerge (avg)** | 27.4 | 60.4 | 27.2 | 60.5 | 7.9 | 60.2 | 6.2 | 60.0 | 20.5 | 59.9 | 22.6 | 60.5 | 47.2 | 60.6 | 7.2 | 60.9 |

Table C2 shows that larger zero-out values with uniformly merged sparsified task vectors lead to improved unlearning results. TIES-merging and MagMax exhibit fewer zero-out values, and their performance is expected to be outperformed by `NegMerge`.

*Table C2.* **Ratio of Zeroed Elements and the Unlearning Performance.** The table reports the zero ratio and the accuracy of the forget set (Acc $D_f$), comparing these values across several baseline methods. Results are obtained using ViT-B/32 Task Arithmetic. Accuracy on $D_r$ is omitted as all methods remain around 60% accuracy on the retain set.

| Method | Cars | | DTD | | EuroSAT | | GTSRB | | MNIST | | RESISC45 | | SUN397 | | SVHN | |
|---|---|---|---|---|---|---|---|---|---|---|---|---|---|---|---|---|
| | % | $D_f(\downarrow)$ | % | $D_f(\downarrow)$ | % | $D_f(\downarrow)$ | % | $D_f(\downarrow)$ | % | $D_f(\downarrow)$ | % | $D_f(\downarrow)$ | % | $D_f(\downarrow)$ | % | $D_f(\downarrow)$ |
| Task Arithmetic[†] | 47.55 | 29.00 | 47.55 | 30.42 | 47.55 | 10.44 | 47.55 | 9.09 | 47.55 | 21.15 | 47.55 | 30.71 | 47.55 | 50.58 | 47.55 | 7.61 |
| MagMax | 47.55 | 35.59 | 47.55 | 31.86 | 47.55 | 10.51 | 47.55 | 8.42 | 47.55 | 20.14 | 47.55 | 30.69 | 47.55 | 55.36 | 47.55 | 9.33 |
| TIES-Merging | 51.59 | 34.03 | 50.62 | 33.09 | 51.62 | 11.56 | 50.72 | 10.21 | 51.94 | 26.09 | 51.00 | 33.41 | 50.58 | 53.80 | 52.06 | 7.48 |
| **NegMerge (ours)** | 90.34 | 27.40 | 92.94 | 27.18 | 94.05 | 7.85 | 94.76 | 6.20 | 93.96 | 20.50 | 92.86 | 22.61 | 92.20 | 47.19 | 92.40 | 7.18 |

*Table C3.* **Sparsity by Component Type.** The results report the sparsity (%) for the Stem and MLP layers across 30 models trained on the Cars dataset. Std. Dev. denotes variability across layers within each component; not applicable to the single-layer Stem.

| Component | ViT-B/32 | | ViT-L/14 | |
|---|---|---|---|---|
| | % | Std. Dev. | % | Std. Dev. |
| Stem | 98.3 | - | 99.6 | - |
| MLP | 55.1 | 0.3 | 77.6 | 0.2 |

Table C3 examines sparsity across different architectural components. Specifically, we compared the Stem layer (*i.e.*, the patch embedding) and MLP layers (c_fc and c_proj weights/biases).

# D. Results in Diverse Model Pool

## D.1. Robustness on Diverse Model Pool

In the CLIP Unlearning Scenario, we conducted additional experiments to analyze the impact of varying hyperparameters, such as weight decay, learning rates, and label smoothing, as shown in Table D1. Also, in Table D2 we present the effects of modifying training hyperparameters and RandAugment configurations. Furthermore, we evaluated using seven different model pools, which are detailed in Table D3. In the Standard Classifier Unlearning Scenario, we enhanced the diversity of the fine-tuned models by adjusting RandAugment configurations, as summarized in Table D4.

As shown in Table D1–Table D4, unlearning performance improves further compared to the baseline, as expected, with diverse types of hyperparameters. These results highlight that, while the degree of improvement may vary depending on the model pool, using multiple models consistently provides more stable performance gains compared to using a single model.

*Table D1.* **Learning Rate, Weight Decay, and Label Smoothing Configuration Pool on ViT-B/32 Task Arithmetic in CLIP Unlearning Scenario.** Results are obtained by evaluating 16 models created from a pool of configurations using the following hyperparameter settings: learning rates of 1e-4, 5e-5, 1e-5, and 5e-6; weight decay values of 0.01 and 0.1; and label smoothing to 0 and 0.1.

| Method | Cars | | DTD | | SUN397 | |
|---|---|---|---|---|---|---|
| | Acc $D_f(\downarrow)$ | Acc $D_r$ | Acc $D_f(\downarrow)$ | Acc $D_r$ | Acc $D_f(\downarrow)$ | Acc $D_r$ |
| Task Arithmetic[†] | 33.52 | 60.29 | 29.14 | 60.38 | 51.36 | 60.55 |
| **NegMerge (ours)** | 30.33 | 60.16 | 26.43 | 59.95 | 47.94 | 60.33 |

*Table D2.* **RandAugment, Learning Rate, and Weight Decay Configuration Pool on ViT-B/32 Task Arithmetic in CLIP Unlearning Scenario.** Results are obtained by evaluating 8 models created from a pool of configurations using the following hyperparameter settings: RandAugment with $n = 1, 2$ and $m = 1, 5, 10$, learning rates of 1e-5, 5e-6, and 1e-6, and weight decay values of 0.01 and 0.1.

| Method | Cars | | DTD | | EuroSAT | | GTSRB | | MNIST | | RESISC45 | | SUN397 | | SVHN | |
|---|---|---|---|---|---|---|---|---|---|---|---|---|---|---|---|---|
| | $D_f(\downarrow)$ | $D_r$ | $D_f(\downarrow)$ | $D_r$ | $D_f(\downarrow)$ | $D_r$ | $D_f(\downarrow)$ | $D_r$ | $D_f(\downarrow)$ | $D_r$ | $D_f(\downarrow)$ | $D_r$ | $D_f(\downarrow)$ | $D_r$ | $D_f(\downarrow)$ | $D_r$ |
| Task Arithmetic[†] | 28.62 | 60.17 | 28.03 | 60.15 | 7.66 | 60.46 | 5.31 | 60.22 | 14.56 | 60.55 | 27.19 | 60.72 | 51.38 | 60.32 | 6.75 | 61.30 |
| **NegMerge (ours)** | 27.42 | 60.03 | 26.80 | 60.08 | 7.03 | 60.39 | 4.82 | 59.50 | 12.89 | 59.94 | 18.23 | 59.81 | 48.73 | 60.38 | 6.71 | 60.29 |

## D.2. Strategy to Create Variants

NegMerge relies on the knowledge encoded in each task vector. Based on our observations, poorly constructed task vectors, such as those trained with unreasonable weight decay, can result in unreliable knowledge and introduce noise into the process. To mitigate this, we recommend constructing a task vector pool using reasonable hyperparameters, ensuring the vectors are reliable and contribute effectively to unlearning.

Furthermore, our method inherently produces a model with a well-optimized retain loss. This aligns with one of our core assumptions discussed in Section 4.3: the merged model should preserve performance on the retain set. Practitioners can leverage this property by using the retain loss as a guiding signal during the model generation phase, enabling more effective model merging through better monitoring and optimization of retain set performance. While we have not yet explored this idea, we view it as an exciting direction for future research.

*Table D3.* **Average of Seven Different Model Pools.** To evaluate robustness across diverse model pools, we assessed performance on seven different model pools. The results show that leveraging multiple models consistently leads to more stable performance improvements compared to relying on a single model.

| Pool | RandAugment (n, m) | Learning Rates | Weight Decay | Label Smoothing | # |
|---|---|---|---|---|---|
| Pool 1 | $n = 1$–$3, m = 1$–$10$ | - | - | - | 30 |
| Pool 2 | - | $1e$-$4, 1e$-$5, 5e$-$5, 5e$-$6$ | $0.01, 0.1$ | $0, 0.1$ | 16 |
| Pool 3 | - | $1e$-$4, 1e$-$5, 5e$-$5, 5e$-$6$ | $0.01, 0.1$ | - | 8 |
| Pool 4 | - | $1e$-$5, 5e$-$6, 5e$-$5$ | $0.01, 0.1$ | - | 6 |
| Pool 5 | $n = 1, 2, m = 5, 10$ | $1e$-$4, 1e$-$5, 5e$-$5, 5e$-$6$ | $0.01, 0.1$ | $0, 0.1$ | 64 |
| Pool 6 | $n = 1, 2, m = 5, 10$ | $1e$-$4, 1e$-$5, 5e$-$5, 5e$-$6$ | $0.01, 0.1$ | - | 32 |
| Pool 7 | $n = 1, 2, m = 1, 5, 10$ | $1e$-$5, 5e$-$6, 1e$-$6$ | $0.01, 0.1$ | - | 8 |

| Pool | Pool 1 ($D_f \downarrow$) | Pool 2 ($D_f \downarrow$) | Pool 3 ($D_f \downarrow$) | Pool 4 ($D_f \downarrow$) | Pool 5 ($D_f \downarrow$) | Pool 6 ($D_f \downarrow$) | Pool 7 ($D_f \downarrow$) |
|---|---|---|---|---|---|---|---|
| Task Arithmetic[†] | 23.63 | 22.80 | 23.21 | 23.21 | 24.05 | 21.31 | 21.19 |
| **NegMerge (ours)** | 20.76 | 21.69 | 21.56 | 22.23 | 22.65 | 19.37 | 19.08 |

*Table D4.* **RandAugment Configuration Pool on ResNet-18 in Standard Classifier Unlearning Scenario.** Results are obtained by evaluating 5 models created from a pool of configurations using the following hyperparameter settings: RandAugment with $n = 1$ and $m = 1, 2, 3, 4, 5$.

| Method | Acc $D_r(\simeq)$ | Acc $D_f(\simeq)$ | Acc $D_{test}(\simeq)$ | MIA($\simeq$) | Avg. Gap($\downarrow$) |
|---|---|---|---|---|---|
| **Retrain *** | 100.00 | 94.76 | 94.26 | 12.88 | 0.00 |
| Task Arithmetic[†] | 97.79 | 95.88 | 91.31 | 9.58 | 2.40 |
| **NegMerge (ours)** | 97.81 | 95.76 | 91.03 | 10.76 | 2.14 |

# E. Theoretical Analysis

Let $\theta_{ori}$ and $\theta_{ft}$ denote the weights of the pre-trained model and a fine-tuned model, respectively. We have the formulation $\theta_{unlearn} = \theta_{ori} - \lambda\tau_{merged}$, where $\tau_{merged}$ is from Equation (1) - our consensually merged task vector. We argue that achieving larger zero-out values with sparsified consensus editing signals in $\tau_{merged}$ could lead to unlearning performance improvements, based on the following fundamental claims: (1) Weight-wise unanimous consensus merging reduces non-zero values and gives a robust $\tau_{merged}$; the larger zero-out values in $\tau_{merged}$ contribute to stable merging. (2) There exists a stable merged point $\theta^*_{unlearn}$, which enjoys better unlearning results as the number of task vectors $\tau_k$ increases.

As the number of task vectors $\tau_k$ in $\tau_{merged}$ increases, the non-zero values decrease because our consensus operation performs like an AND operation. The robustness of $\tau_{merged}$ increases upon merging, as sparse weights are merged uniformly, which enjoys inherently more robustness than an individual weight as revealed in (Wortsman et al., 2022; Jang et al., 2024), where the first term of $\tau_{merged} = \frac{1}{n} * \sum_{k=1}^{n} \theta_{ft} - \theta_{ori}$ conducts uniform merge. $\theta_{unlearn}$ moves closer to $\theta_{ori}$, which likely stays in lower loss regions due to weights that generally hold linear mode connectivity (LMC) (Frankle et al., 2020; Juneja et al., 2022; Entezari et al., 2021). It also mitigates issues that cause fluctuating high losses (i.e., loss barriers) on certain loss surfaces, when weights deviate significantly from $\theta_{ori}$. Therefore, task negation at an improved $\theta^*_{unlearn}$ is likely to reside in lower loss regions, leading to better results. While we do not specify the exact $\theta^*_{unlearn}$, increasing the number of merged task vectors allows the process to approach a closer-to-optimal point as more sparsified merged weights are merged uniformly (Jang et al., 2024).

Our empirical evidence, shown in Table 4 demonstrates that larger zero-out values from uniformly merged sparsified task vectors lead to improved unlearning results. Additionally, as indicated in Table C2, TIES-merging and MagMax exhibit fewer zero-out values, which explains their lower performance compared to our method.

# F. Applicability Beyond Image Classification

While our study focuses on image classification tasks, we believe NegMerge is broadly applicable to other modalities, including large language models (LLMs) and vision-language models (VLMs). Our approach leverages the consistency of directional changes in parameter space across multiple fine-tuned models, a principle not specific to visual tasks or

architectures. Given that LLMs and VLMs are often built on transformer backbones, which our method already demonstrates strong performance on (*e.g.*, ViT-B/32, ViT-L/14), we expect the sign-consensual merging strategy to extend to these domains.

