# OpenReview forum: "NegMerge: Sign-Consensual Weight Merging for Machine Unlearning"
_ICML.cc/2025/Conference — ICML 2025 poster_

### Official Review · Reviewer_hSw6 · 2025-02-19

**Overall Recommendation:** 3

**Summary:**

This paper presents NegMerge, which enhances the process of forgetting by negation for machine unlearning. NegMerge computes a final task vector by merging task vectors derived from multiple fine-tuned models, during which it preserves elements with consistent signs across the task vectors and masks those with inconsistent signs to zero. The results demonstrate that NegMerge achieves state-of-the-art performance.

**Claims And Evidence:**

The paper claims that the effectiveness of current methods is highly sensitive to hyperparameter selection, which is presented in Figure 1.

**Essential References Not Discussed:**

The related work described in this paper primarily encompasses model merging and machine unlearning.

**Experimental Designs Or Analyses:**

Yes.

**Methods And Evaluation Criteria:**

Yes, it follows the previous literature.

**Other Comments Or Suggestions:**

none.

**Other Strengths And Weaknesses:**

1. One of the motivations of this paper is that "hyperparameter sets that preserve retain set performance tend to exhibit poor unlearning performance, and vice versa." The reasoning behind why this motivation leads to the use of model merging to address issues in machine unlearning is not clearly explained. In other words, it is not well-articulated why model merging techniques are particularly suitable for the direction of machine unlearning.

2. The issue that "unlearning performance is highly sensitive to the hyperparameter settings used for obtaining fine-tuned models" is presumably widespread in the field of machine unlearning. How does this paper mitigate this phenomenon, or which aspect of model merging technology alleviates this issue?

3. In Table 2, several metrics including Acc D_r, Acc D_f, and MIA are presented. It is not clearly stated whether higher values are better or lower values are preferable for these indicators.

4. In Table 3, the "Conflict" ablation version is inferior to the "All" version. Does this suggest that the method of selecting parameters using conflicting signs is not very effective?

**Questions For Authors:**

Please see the "Other Strengths And Weaknesses" part.

**Relation To Broader Scientific Literature:**

The primary contribution of this paper lies in the application of model merging techniques to the field of machine unlearning.

**Theoretical Claims:**

In Section E, Theoretical Analysis, the paper provides a partial theoretical analysis.

---

> ### Author Rebuttal · Authors · 2025-04-01
>
> We thank the reviewer for the constructive questions. We have done our best to address the concerns raised, as detailed below.
>
> ### **Weakness 1: Why Model Merging for Machine Unlearning?**
>
> This is because models fine-tuned with different hyperparameters tend to specialize in either unlearning or retaining  (as shown in Figure 1). Therefore, while searching for effective hyperparameters is a prerequisite for approaching optimality, we argue that merging them provides a way to combine their strengths and resolve the trade-off that a single model cannot, which also holds in unlearning. As shown in Figure 1(a), no single model achieves high performance on both objectives.
>
> We believe model merging is effective for unlearning because it provides access to multiple optimization trajectories, which allows us to identify parameters that are related to the forget set (e.g., via sign consensus). This information is not available from a single model. By selectively negating these parameters while preserving others, we can more precisely remove unwanted information without degrading performance on the retain set. As shown in Table 1, the merged model consistently outperforms all single models. We will clarify this point in the final paper.
>
> ### **Weakness 2: Mitigating Hyperparameter Sensitivity via Merging**
>
> Due to the sensitivity issue, obtaining a model with strong unlearning performance inevitably requires a hyperparameter search process. However, rather than selecting a single best model from this search, we mitigate the sensitivity by merging all fine-tuned models obtained from diverse hyperparameter configurations. This strategy is inspired by the model soups framework (Wortsman et al., 2022), which shows that averaging the weights of independently fine-tuned models can improve performance and robustness. The key idea is that different hyperparameter choices introduce independent variations, and merging helps cancel out noise from any single run. As a result, the final model is less sensitive to specific hyperparameter choices, while still capturing the shared structure needed for effective unlearning. We will highlight this point in the final paper.
>
> ### **Weakness 3: Clarifying Metrics in Table 2 (Acc Dr, Acc Df, MIA, etc.)**
>
> Each metric (Acc Dr, Acc Df, Acc Dtest, MIA) is favorable when its value is equal to that of the Retrain model (“≃”), indicating that the ideal Unlearn model needs to behave similarly to the Retrain model. This notation will be clarified in the caption of Table 2.
>
> ### **Weakness 4: The “Conflict” Ablation in Table 3**
>
> The weaker performance of the "Conflict" method suggests that using conflicting-sign parameters is not effective. As shown in Section 4.3, it degrades unlearning performance. In contrast, the "Consensus" method, which uses only sign-consistent elements, performs best, indicating that consistent signs better capture changes tied to the forget set, while conflicting signs likely reflect noise from different training setups.

---

### Official Review · Reviewer_xEch · 2025-03-14

**Overall Recommendation:** 3

**Summary:**

The paper proposes a novel framework for machine unlearning that takes advantage of multiple fine-tuned models (10 to 30) to localize the parameters that have the same sign across all task vectors. The idea is that these parameters are characteristic of the task, while others might be noise. The procedure results in a new task vector which is then subtracted from the zeroshot model. The authors include results on CLIP unlearning and on CIFAR with ResNets.

**Claims And Evidence:**

The claims of the paper are clear and convincing.

**Essential References Not Discussed:**

* Machine unlearning of LLMs is not discussed
* See Weaknesses for some more references.

**Experimental Designs Or Analyses:**

The experimental design is sound. The “Avg gap” metric is not clear from the text.

**Methods And Evaluation Criteria:**

The proposed methods and evaluation criteria do make sense for the problem at hand. The evaluation is thorough and well documented on the selected benchmarks. The paper, however, does omit experiments on machine unlearning of LLMs, which is an important topic.

A minor issue is the lack of retain performance on table 4.

**Other Comments Or Suggestions:**

1. See weaknesses
2. It would be useful to include a graph of consensus as a number of available checkpoints. This can be done per layer, since previous works have shown that different types of layers (e.g. attention vs mlp layers) have different levels of importance. For x models, there are (30 choose x) combinations, hence subsampling could work.
3. NTK reference is Jacot et al. The cited one Ortiz-Jimenez et al applies NTK ideas in task arithmetic. Please fix.

**Other Strengths And Weaknesses:**

## Strengths

* The idea of leveraging multiple fine-tuned models in machine unlearning is original. The core of the idea has been seen before, e.g., model soups etc, but afaik this is the first work to apply these concepts to machine unlearning.
* The paper is easy to follow and well written
* The experimental validation is thorough, ablations on the number of models are done (see also below for suggestion)

## Weaknesses

1. Lack of LLM experiments. Machine unlearning for language is an important aspect of the literature (if not the most important) and is not discussed or used in experiments
2. The paper could be further improved if more intuition is provided. Specifically, L146-158 (step 2\) provides some intuition for why the proposed method works but it is very limited. Imo, the method is interesting because it reveals that certain directions are characteristic of the task and hence appear in all models regardless of configurations (as mentioned in the paper), while everything else is noise. Hence, connecting to literature such as \[1,2\]  would be very interesting. Similarly, the proposed method performs some kind of task localization (especially in the task addition section) and, therefore, works on this area should also be cited \[3,4,5\]. I believe more intuition and connecting to these areas would improve the paper.

\[1\] Morcos, Ari, Maithra Raghu, and Samy Bengio. "Insights on representational similarity in neural networks with canonical correlation." Advances in neural information processing systems 31 (2018).

\[2\] Kornblith, Simon, et al. "Similarity of neural network representations revisited." International conference on machine learning. PMLR, 2019\.

\[3\] Wang, Ke, et al. "Localizing task information for improved model merging and compression." arXiv preprint arXiv:2405.07813 (2024).

\[4\] He, Yifei, et al. "Localize-and-stitch: Efficient model merging via sparse task arithmetic." arXiv preprint arXiv:2408.13656 (2024).

\[5\] Panigrahi, Abhishek, et al. "Task-specific skill localization in fine-tuned language models." International Conference on Machine Learning. PMLR, 2023\.

**Questions For Authors:**

See above.

**Relation To Broader Scientific Literature:**

The paper misses some related works that can better place the work in the broader scientific literature. See weaknesses.

**Theoretical Claims:**

There are no formal theoretical claims. The paper does include an appendix about a theoretical analysis but it is more about intuition rather than formal claims.

---

> ### Author Rebuttal · Authors · 2025-04-01
>
> We thank the reviewer for recognizing our method’s originality. Below, we address concerns on LLM experiments and related work.
>
> ### **Weakness 1: Lack of LLM experiments**
>
> We agree that unlearning in LLMs is an important direction. We believe our method can generalize to transformer-based LLMs given its demonstrated effectiveness on transformer architectures, although the current study focuses on image classification tasks. We appreciate the suggestion and will include LLM experiments and discussion on our method’s applicability to LLMs in the final version of the paper.
>
> ### **Weakness 2: Additional references and deeper intuition (CCA, task localization)**
>
> #### **1. Connection to Canonical Correlation Analysis (CCA)**
>
> As the reviewer suggested, CCA-based methods [1,2] have been used to analyze similarity between layers of neural networks. Indeed, it would be interesting to adapt such a viewpoint to identify which layers are most effective for unlearning. Our work, however, starts from the assumption that certain parameters—rather than entire layers—are more central to the given task. We thus focus on parameter-level unlearning, where we isolate specific parameters that are consistently implicated in the task across multiple models. We agree that layer-level unlearning via CCA is a promising direction, and will add discussion.
>
> #### **2. Relation to Task Localization / Merging Methods**
>
> We appreciate the additional pointers to task localization methods [3,4,5]. In our Related Work section, we cite some approaches (e.g., TIES-Merging, AdaMerging, MagMax) that also address how to combine or localize tasks. However, these methods focus largely on task addition, attempting to merge models trained on different tasks to maximize the differences that define each new task. For instance, the Tall Mask method [3] seeks “taller” vectors for merging.
>
> In contrast, our goal is to identify parameters shared across models trained on similar tasks. Rather than negating the most diverse features, we look for the parameters that share a consistent effect (sign) across models. These shared parameters most effectively capture the task vectors that are attributed to the forget set and are therefore the prime candidates for unlearning. We will further clarify this difference in the revised paper.
>
> ### **Suggestions 1: Lack of retain set (Dr) performance in Table 4**
>
> We have added the full Table 4 results (please see Reviewer aohJ, Weakness 4) and will clarify further in the final version.
>
> ### **Suggestions 2: “Avg gap” metric not clearly defined**
>
> Following SalUn’s evaluation protocol, we report the Avg. Gap, which is the average difference between the Unlearn and Retrain models across Dr, Df, Dtest, and MIA. We will add further clarification in the final version.
>
> ### **Suggestions 3: Sign consensus analysis**
>
> We conducted several analyses using ViT-B/32 and ViT-L/14 on the Cars dataset with 30 models. First, we examined the block-wise consensus ratio. As shown in the tables below, the consensus ratio increases in deeper blocks, in both architectures. This suggests that more parameters are being negated as the blocks go deeper. Here, we grouped blocks into three ranges and reported the average consensus ratio per range. Each block corresponds to one transformer block.
>
> **ViT-B/32:**
> | Block Range | Consensus Ratio |
> |-------------|------------------|
> | 0–3         | 76.55           |
> | 4–7         | 81.64           |
> | 8–11        | 87.08           |
>
>
> **ViT-L/14:**
> | Block Range | Consensus Ratio |
> |-------------|------------------|
> | 0–7         | 66.52           |
> | 8–15        | 73.62           |
> | 16–23       | 76.68           |
>
> Additionally, we compared the Stem layer (i.e., the patch embedding), MLP layers (c_fc and c_proj weights/biases), and Attention layers (query, key, value, and output projection weights/biases). The standard deviation refers to the variability across different layers within each category (MLP or attention). Surprisingly, the Attention layers consistently showed 100% sign consensus in both architectures. We hypothesize that this is because the query-key-value mechanism plays a fundamental role in capturing contextual relationships, which remain stable across different fine-tuning configurations.
>
> **ViT-B/32:**
> | Layer     | Consensus (%) | Std Dev |
> |-----------|----------------|---------|
> | Stem      | 1.75           | 0.00    |
> | Attention | 100.00         | 0.00    |
> | MLP       | 44.87          | 0.31    |
>
> **ViT-L/14:**
> | Layer     | Consensus (%) | Std Dev |
> |-----------|----------------|---------|
> | Stem      | 0.36           | 0.00    |
> | Attention | 100.00         | 0.00    |
> | MLP       | 22.41          | 0.23    |
>
> We will include these findings, which we believe enhance the paper’s depth and quality. We sincerely thank the reviewer for the helpful suggestion that led to this.
>
> ### **Suggestions 4: NTK reference**
>
> We will ensure that our next revision will correct it.

---

> > ### Comment · Reviewer_xEch · 2025-04-08
> >
> > Thank you for your rebuttal. The sign consensus analysis per layer is really interesting and can be featured in the paper. Since your analysis reveals that layers become progressively more important wrt depth, works that have also focused on this aspect (in general such as [6, 7] or model merging specific) should be included as references.
> >
> > Regarding your comment on **Weakness 2**:
> >
> > * My point on the CCA works was not for the authors to create a new baseline where the entire layers are removed/modified towards unlearning, but merely to connect to these works for more intuition. One of my major concerns with this paper is lack of intuition and, therefore, citing relative literature can help.
> >
> > * I disagree with your comment on task localization. First, TIES does not "localize" any tasks; each parameter is treated individually and its final value only depends on the values for the same parameter coming from the task vectors. In other words, for T task vectors, there are T values per parameter and the final value depends on these T scalars. Second, Ada-Merging also does not localize any tasks; rather it optimizes the scalings per task and per layer towards *a single objective* which is minimizing the average across tasks loss. Finally, and more importantly, the sentence *"the Tall Mask method [3] seeks “taller” vectors for merging."* does not make any sense - what are "taller" vectors?
> >
> > Despite the difference in settings (addition vs negation, multiple fine-tuned tasks on same rather than different models), [3,4,5] are very relevant to your work since they identify a subset of parameters important to each task (e.g. param 1 from layer 2 *and* param 3 from layer 5 are important for task 1 etc).
> >
> > [6] Yosinski, Jason, Jeff Clune, Yoshua Bengio, and Hod Lipson. "How transferable are features in deep neural networks?." Advances in neural information processing systems 27 (2014).
> >
> > [7] Neyshabur, Behnam, Hanie Sedghi, and Chiyuan Zhang. "What is being transferred in transfer learning?." Advances in neural information processing systems 33 (2020): 512-523.

---

> > > ### Author Response · Authors · 2025-04-09
> > >
> > > ### **1. Analysis per Layer**
> > >
> > > Thank you for the insightful suggestion. We agree that understanding layer-wise behavior is crucial, and appreciate the pointers to [6,7].
> > >
> > > Interestingly, our sign-consensus analysis reveals that deeper layers undergo more negation, even though our method wasn’t explicitly designed to be layer-aware. This aligns with prior observations: [6] and [7] show that lower layers capture general features while deeper ones specialize, and are more sensitive to perturbations.
> > >
> > > We also found a related result in [8]—they observe that shallow layers are more affected by unlearning, as they encode general knowledge. While they don't perform layer-wise unlearning, they suggest it as a future direction.
> > >
> > > We’ll incorporate this discussion into the paper—thank you for helping us make this connection clearer.
> > >
> > > [8] Wang, Qizhou, et al. "Rethinking LLM Unlearning Objectives: A Gradient Perspective and Go Beyond." The Thirteenth International Conference on Learning Representations.
> > >
> > > ### **2. CCA**
> > >
> > > Thank you for the clarification. We agree that citing CCA-based works [1,2] can help clarify the intuition behind our approach and will update the paper accordingly.
> > >
> > > ### **3. Task Localization**
> > >
> > > Thank you for your valuable feedback. As you rightly pointed out, TIES and Ada-Merging are not parameter localization methods, and we acknowledge that our previous wording may have been confusing. We will revise the corresponding parts to clarify this point.
> > >
> > > Our method and [3,4,5] are both parameter localization methods, with different strategies. For example, [3] masks out parameters with small magnitudes, [4] (Dataless Localization) retains the top-k% parameters based on magnitude, and [5] selects the top-s parameters showing the largest changes during fine-tuning. In contrast, we identify parameters whose directionality is consistent across models trained on the same dataset, thereby localizing those most associated with the forget set.
> > >
> > > We agree that [3,4,5] are highly relevant, and we will cite them accordingly.
> > >
> > > Regarding the phrase “taller vectors,” we intended to refer to components with larger absolute values that are retained through masking. We agree that the term was unclear and will revise or remove it to avoid confusion.

---

### Official Review · Reviewer_zcBE · 2025-03-15

**Overall Recommendation:** 2

**Summary:**

This paper proposes a new approach to machine unlearning focusing on instance-based unlearning, where instead of forgetting samples corresponding to a specific class, it can forget samples throughout all the classes while preserving model performance on the rest of the samples, i.e., retain-set.

**Claims And Evidence:**

Yes.

**Essential References Not Discussed:**

N/A.

**Experimental Designs Or Analyses:**

Yes. The proposed approach in Section~3.2.

**Methods And Evaluation Criteria:**

Yes.

**Other Comments Or Suggestions:**

N/A.

**Other Strengths And Weaknesses:**

**Strenghts**

1. The paper is well written, and the problem setup is clear.
2. The proposed approach could provide an alternative to the important parameter-based unlearning approaches, which suffer from performance degradation on the retain-set samples while enabling unlearning on the forget-set samples.

**Weaknesses**

1. One of the key weaknesses of this paper is limited evaluation. The authors do not provide any empirical evaluation on the class unlearning setup. Furthermore, the authors only provide instance unlearning on CIFAR10 with Resnet. I believe it is crucial to include comparisons involving various datasets, CIFAR100, ImageNet-1K, and TinyImageNet is crucial. Moreover, the comparison involving different vit architecture involving Swin transformer, ViT-B/16, ViT-L/16 using aforementioned datasets is also necessary.

2. In Table~2, if we check the column Acc $(D_{f})$, it is almost similar to Acc $(D_{r})$, which raises a crucial question: is this approach really enabling instance unlearning?

**Questions For Authors:**

Please refer to the weaknesses section.

**Relation To Broader Scientific Literature:**

The proposed approach could provide an alternative to the important parameter-based unlearning approach.

**Theoretical Claims:**

No.

---

> ### Author Rebuttal · Authors · 2025-04-01
>
> We appreciate the reviewer’s positive comments on the clarity of our paper and for recognizing the potential of our method as an alternative to parameter-based unlearning approaches. The reviewer has expressed concerns regarding the generalizability of our work and questions regarding instance-wise unlearning. We hope the following feedback addresses these concerns.
>
> ### **Weakness 1: Limited evaluation: class unlearning, more datasets, and different architectures**
>
> We provide additional results for class-wise unlearning and 50% random data forgetting on CIFAR-10 using ResNet18, using only the forget set. As shown below, our method consistently surpasses baseline methods. Please refer to our response to Reviewer aohJ (Weakness 3) for results on the 50% random forgetting setting.
>
> **Class-wise Unlearning Results:**
>
> | Method                 | Dr    | Df    | Dtest | MIA   | Avg.Gap ↓ |
> |------------------------|-------|-------|-------|-------|-----------|
> | Retrain                | 100.0 | 0.0   | 92.5  | 100.0 | 0.0       |
> | Random Labeling        | 83.0  | 10.1  | 70.9  | 99.5  | 12.3      |
> | SalUn                  | 86.5  | 10.8  | 74.1  | 100.0 | 10.7      |
> | Task Arithmetic (Best) | 95.3  | 0.1   | 80.6  | 100.0 | 4.2       |
> | Ours                   | 96.8  | 0.8   | 81.8  | 99.8  | 3.7       |
>
> Additionally, we present new results on 10% random data forgetting conducted on TinyImageNet and Swin-T. With this, we now evaluate instance unlearning across three datasets—CIFAR-10, TinyImageNet, and CUB—using three architectures: ResNet-18, VGG-16, and Swin-T. Our approach consistently outperforms baseline methods across various datasets and architectures, including TinyImageNet and Swin-T.  Please find our new results below.
>
> On the **TinyImageNet** dataset:
> | Method                 | Dr    | Df    | Dtest | MIA  | Avg.Gap ↓ |
> |------------------------|-------|-------|-------|------|-----------|
> | Retrain                | 100.0 | 63.6  | 63.7  | 63.8 | 0.0       |
> | Random Labeling        | 76.4  | 76.1  | 58.1  | 32.3 | 18.3      |
> | SalUn                  | 73.6  | 73.8  | 56.7  | 30.6 | 19.2      |
> | Task Arithmetic (Best) | 76.9  | 73.6  | 59.4  | 29.3 | 18.0      |
> | Ours                   | 76.0  | 71.9  | 58.6  | 31.3 | 17.5      |
>
> Using **Swin-T**:
> | Method                 | Dr   | Df   | Dtest | MIA | Avg. Gap ↓ |
> |------------------------|------|------|-------|-----|-------------|
> | Retrain                | 100.0| 97.8 | 97.7  | 4.7 | 0.0         |
> | Random Labeling        | 100.0| 100.0| 97.7  | 0.6 | 1.6         |
> | SalUn                  | 99.0 | 99.0 | 96.4  | 3.4 | 1.2         |
> | Task Arithmetic (Best) | 98.5 | 97.8 | 96.0  | 4.0 | 1.0         |
> | Ours                   | 98.8 | 97.8 | 95.9  | 4.6 | 0.8         |
>
> ### **Weakness 2: Is instance unlearning truly happening?**
>
> In the standard classifier unlearning scenario, the Retrain model is trained from scratch using only the retain set and excluding the forget set, and has been regarded as the (ground-truth-like) upper bound of performance, as none of the forget set is used during training. The goal is for the Unlearn model to closely match the Retrain model's performance on each of Dr, Df, Dtest, and MIA individually, which has been the standard evaluation protocol in this scenario. In instance-level unlearning, classification accuracy on the forget set often remains similar, as observed in the corresponding table. This is presumably due to the generalization effect: even though the Retrain model is not trained on the forget set, it can still perform well on it by leveraging class-level information learned from the retain set.

---

> > ### Comment · Reviewer_zcBE · 2025-04-05
> >
> > Thank you for the rebuttal. However, it only partially addresses my concerns.
> >
> > **Class-wise unlearning** - It can be seen that the proposed approach achieves really good forget-set accuracy. However, the MIA score is really, so one can simply argue that the proposed model is not actually forgetting or unlearning properly. In case of instance unlearning I may have considered that it might be the effect of generalization. However, with class unlearning that same justification will not work and MIA should also come down along with the forget set accuracy. Also for retrain model, MIA should be less, and not 100 [1].
> >
> > [1] Foster, Jack, Stefan Schoepf, and Alexandra Brintrup. "Fast machine unlearning without retraining through selective synaptic dampening." In Proceedings of the AAAI conference on artificial intelligence, vol. 38, no. 11, pp. 12043-12051. 2024.

---

> > > ### Author Response · Authors · 2025-04-05
> > >
> > > Thank you for your comment. Our MIA evaluation follows Salun [2], which adopted the MIA-Efficacy metric from the paper - Model Sparsity Can Simplify Machine Unlearning [3]. Notably, this metric is not an attack success rate but rather similar to an attack failure rate. Higher MIA-Efficacy implies better unlearning, as it measures how much less information the model retains about the forget data. For details, please refer to Appendix C.3 of [3]. Notably, both [2, Table A2] and [3, Figure 5] report MIA-Efficiency scores close to 100 in the class-wise unlearning setting.
> > >
> > > We also acknowledge the reviewer’s point and have re-evaluated MIA using the logistic regression-based MIA, similarly following the suggested paper Fast Machine Unlearning [1] (please see the rightmost column in the table below). The updated  MIA values are significantly lower and better align with the intuition that unlearning has been successful. We will include the updated results and relevant details in the revised paper.
> > >
> > > **Class-wise Unlearning Results:**
> > >
> > > | Method                 | Dr    | Df    | Dtest | MIA [2] | MIA [1]
> > > |------------------------|-------|-------|-------|-------|-------|
> > > | Retrain                | 100.0 | 0.0   | 92.5  | 100.0 | 0.00       |
> > > | Random Labeling        | 83.0  | 10.1  | 70.9  | 99.5  |0.04
> > > | SalUn                  | 86.5  | 10.8  | 74.1  | 100.0 |0.00
> > > | Task Arithmetic (Best) | 95.3  | 0.1   | 80.6  | 100.0 |0.02
> > > | Ours                   | 96.8  | 0.8   | 81.8  | 99.8  |0.16
> > >
> > > [1] Foster, Jack, Stefan Schoepf, and Alexandra Brintrup. "Fast machine unlearning without retraining through selective synaptic dampening." In Proceedings of the AAAI conference on artificial intelligence, vol. 38, no. 11, pp. 12043-12051. 2024.
> > >
> > > [2] Fan, Chongyu, et al. "SalUn: Empowering Machine Unlearning via Gradient-based Weight Saliency in Both Image Classification and Generation." The Twelfth International Conference on Learning Representations.
> > >
> > > [3] Jia, Jinghan, et al. "Model sparsity can simplify machine unlearning." Advances in Neural Information Processing Systems 36 (2023): 51584-51605.

---

### Official Review · Reviewer_aohJ · 2025-03-15

**Overall Recommendation:** 2

**Summary:**

This paper treats the unlearning problem as a task arithmetic, where they conduct task vector by finetuning on the forget set, then subtract it from the original weight. To avoid sensitivity to hyperparameter selection, the authors create a finetuned model pool by various hyperparameter settings and aggregate them into a final task vector by considering their signs. They claim that the proposed method requires similar or fewer computational complexity than existing methods, and achieves superior performance compared to state-of-the-art methods.

**Claims And Evidence:**

The evidence is not enough to convince the effectiveness of the proposed method. When compared with existing unlearning methods, the authors only conducted an experiment on 10% random data forgetting on CIFAR-10. However, SALUN [1] has demonstrated its effectiveness on <10,20,30,40,50>% Random Data Forgetting on CIFAR-10, CIFAR-100, and Tiny-Imagenet, whereas Boundary Unlearning [2] has demonstrated its effectiveness on class-wise unlearning setting.

[1] Fan, Chongyu et al. “SalUn: Empowering Machine Unlearning via Gradient-based Weight Saliency in Both Image Classification and Generation.” ArXiv abs/2310.12508 (2023): n. pag.
[2] Chen, Min et al. “Boundary Unlearning: Rapid Forgetting of Deep Networks via Shifting the Decision Boundary.” 2023 IEEE/CVF Conference on Computer Vision and Pattern Recognition (CVPR) (2023): 7766-7775.

**Essential References Not Discussed:**

No essential reference is not discussed.

**Experimental Designs Or Analyses:**

The baselines are verified to be effective on class-wise unlearning and various scenarios of instance-wise unlearning, however, the authors did not conduct sufficient experiment settings, making the comparison less convincing.

**Methods And Evaluation Criteria:**

The proposed methods are reasonable for unlearning. Also, evaluation criteria are the most popular benchmark and metrics, which are used in current state-of-the-art models.

**Other Comments Or Suggestions:**

There are a few typos can be improved.

**Other Strengths And Weaknesses:**

Strengths:

-	Unlearning is an important problem in practice.
-	The proposed methods are intuitive and well-presented.
-	The paper is well-written and easy to follow.

Weaknesses:

-	No theoretical claims were provided to understand the effectiveness of the proposed method rigorously.
-	The inability to conduct experiments on ViT-L/14 raises concerns about the feasibility of applying these to large models and real-world applications.
-	The experiments are weak to make the claims. When compared to existing unlearning methods, the authors only conduct experiments on the CIFAR10 dataset with the configuration of 10% Random Forgetting. It should be done on 50% random forgetting and class-wise unlearning, where current methods have demonstrated their effectiveness in those settings.
-	In the study of sparsity, only performance on the forget set has been reported. The performance on retain set is also important, so it should not be ignored from the experiment analysis.

**Questions For Authors:**

-	Is the proposed method efficient when the pool size is large?

**Relation To Broader Scientific Literature:**

Sign consistency is an interesting approach to understand the contribution of each parameter in a model. It is feasible to apply to model pruning, model compression, model editing, etc.

**Theoretical Claims:**

No theoretical claims.

---

> ### Author Rebuttal · Authors · 2025-04-01
>
> We appreciate the reviewer’s positive feedback on the clarity and intuitiveness of our method. We hope to address the concerns regarding generalizability and scalability in our responses below.
>
> ### **Weakness 1: No theoretical claims were provided**
>
> We provide a theoretical claim for our method. We would like to note that an informal version was presented in Appendix E.
>
> **Theorem.** Consensus-based merging of multiple task vectors yields a robust and effective unlearning direction by inducing sparsity and guiding the model toward a low-loss region.
>
> **Lemma 1.** As the number of task vectors $\tau_k$ increases, the merged vector $\tau_{merged}$ becomes sparser.
>
> **Proof.** Consensus merging keeps each element only if all task vectors agree in sign, acting like an AND operation. With more vectors, agreement decreases, increasing sparsity.
>
> **Lemma 2.** If $\theta\_{ft}$ is centered around $\theta\_{ori}$, then $\theta^*\_{unlearn} = \theta\_{ori} - \tau\_{merged}$ lies closer to $\theta\_{ori}$.
>
> **Proof.** Given $\tau\_{merged} = \bar{\theta}_{ft} - \theta\_{ori}$,
>
> we have $\theta^*\_{unlearn} = \theta\_{ori} - (\bar{\theta\}_{ft} - \theta\_{ori})$.
>
> If $\bar{\theta}_{ft} \approx \theta\_{ori}$, then:  $\theta^*\_{unlearn} \approx  \theta\_{ori}$.
>
> Lemmas 1 and 2 imply that consensus merging improves robustness (via sparsity) and effectiveness (by staying near $\theta_{ori}$ under linear mode connectivity), yielding a reliable unlearning direction.
>
>
> While our theoretical analysis may offer a reasonably plausible justification, its effectiveness was most clearly demonstrated through strong empirical results across diverse datasets and architectures. We consider a more rigorous theoretical analysis as future work.
>
> ### **Weakness 2: Inability to conduct experiments on ViT-L/14**
>
> ViT-L/14 results for Task Arithmetic were actually included in Table 1. For Linear Task Arithmetic, we ran a scaled-down experiment on ViT-L/14 using six models on the Cars dataset to complete it within the rebuttal period. As shown in the table below, ours outperforms the baselines, achieving the lowest forget accuracy (Df ↓) while maintaining comparable retain accuracy (Dr).
>
> | Method             | Df ↓  | Dr    |
> |--------------------|-------|-------|
> | Single Best Model  | 28.68 | 71.45 |
> | Uniform Merge      | 34.68 | 71.96 |
> | Ours               | 24.47 | 71.66 |
>
>
> ### **Weakness 3: Evidence is not enough (50% random forgetting and class-wise unlearning)**
>
> We extend Table 2 with additional experimental results under 50% random data forgetting and class-wise unlearning, using only the forget set. These experiments were conducted on the CIFAR-10 dataset with the ResNet18 architecture. As shown below, our method outperforms baselines in unlearning performance. For class-wise unlearning and 10% forgetting on TinyImageNet and Swin-T, please see our response to Reviewer zcBE (Weakness 1).
>
> **50% Random Data Forgetting Results:**
>
> | Method                 | Dr    | Df    | Dtest | MIA  | Avg.Gap ↓ |
> |------------------------|-------|-------|-------|------|-----------|
> | Retrain                | 100.0 | 92.1  | 91.7  | 19.3 | 0.0       |
> | Random Labeling        | 99.8  | 99.9  | 94.7  | 2.2  | 7.0       |
> | SalUn                  | 99.6  | 99.6  | 94.2  | 4.4  | 6.3       |
> | Task Arithmetic (Best) | 98.4  | 97.9  | 92.6  | 5.6  | 5.5       |
> | Ours                   | 96.8  | 96.5  | 91.5  | 6.3  | 5.2       |
>
> ### **Weakness 4: Lack of retain set (Dr) performance in Table 4**
>
> In Section 4.2, we mentioned that the retain set (Dr) accuracies for all methods remain around 60%, following the implementation details of Ilharco et al. (2022) for the CLIP unlearning scenario. Below is the extended table (Table 4) including both Df and Dr results. We will provide further clarification in the final version.
>
> | #   | Cars (Df↓ / Dr↑) | DTD (Df↓ / Dr↑) | SUN397 (Df↓ / Dr↑) |
> |-----|------------------|------------------|---------------------|
> | 5   | 30.5 / 60.4      | 28.8 / 60.5      | 49.7 / 60.5         |
> | 10  | 26.6 / 59.9      | 27.8 / 60.4      | 48.7 / 60.6         |
> | 15  | 26.0 / 60.0      | 27.7 / 60.4      | 47.7 / 60.5         |
> | 20  | 26.1 / 60.1      | 27.1 / 60.4      | 47.8 / 60.6         |
> | 25  | 26.6 / 60.2      | 27.3 / 60.5      | 47.0 / 60.4         |
> | 30  | 27.4 / 60.4      | 27.2 / 60.5      | 47.2 / 60.6         |
>
>
> ### **Suggestions 1: Minor typos**
>
> We appreciate the reviewer’s notice. We will carefully proofread again before the final submission.
>
> ### **Questions 1: Is the proposed method efficient when the pool size is large?**
>
> Although our method scales with the number of fine-tuned models, it does not incur additional overhead, since all methods—including single-task arithmetic—share the same underlying model pool. Comparable cost increases also arise in single-task arithmetic due to its broader hyperparameter search space. This is discussed in detail in Section 3.3, Analyses on Computational Cost.

---

### Decision · Program_Chairs · 2025-05-01

**Decision:**

Accept (poster)

**Comment:**

This paper proposes an unlearning method based on merging task arithmetic vectors. The idea is to merge task vectors obtained from various hyperparameters, instead of using a hyperparameter search process, and by that to reduce the computational load of unlearning. The merging is based on the signs of the task vector components, where only components with sign agreement are used for the merged task vector.
The paper shows that the proposed unlearning method performs well for both vision-language models such as CLIP ViT and for image classification.

Following comments by Reviewers zcBE and aohJ, the authors added in their rebuttal results for 50% random forgetting, class-wise unlearning, TinyImageNet, some results for ViT-L/14, and Swin transformers. This makes the experimental evaluation of the proposed method to be sufficiently extensive.

It is worth to mention here the following two weaknesses:
* The authors' theoretical claim is not rigorous or extensive enough, as pointed out in the review. However, while it is always good to have some theory, it is not strictly necessary in a paper that proposes a new unlearning method based on a sound idea and showcases its performance empirically.
* Reviewer zcBE wrote that having a forget accuracy closer to the retain accuracy than to the test accuracy is a weakness of the proposed unlearning method, even if the unlearning method does not use the forget set samples. Table 2 shows that there are many other unlearning methods that have this weakness, although there might be exceptions like the method in the paper mentioned by Reviewer zcBE. While this is indeed a weakness to mention, it is not necessarily a reason to not accepting this paper.

In summary, this paper has novelty and strengths that outweigh its weaknesses and my recommendation is to consider it has a candidate for acceptance (weak accept).